# Three-dimensional magnetic nanotextures with high-order vorticity in soft magnetic wireframes

Oleksii M. Volkov [1] ✉, Oleksandr V. Pylypovskyi [1,2] ✉, Fabrizio Porrati [3] ✉, Florian Kronast [4], Jose A. Fernandez-Roldan[1], Attila Kákay [1], Alexander Kuprava[3], Sven Barth [3], Filipp N. Rybakov [5], Olle Eriksson [5,6], Sebastian Lamb-Camarena[7,8], Pavlo Makushko[1], Mohamad-Assaad Mawass [4,9], Shahrukh Shakeel[1], Oleksandr V. Dobrovolskiy [7], Michael Huth [3] & Denys Makarov [1] ✉

Additive nanotechnology enable curvilinear and three-dimensional (3D) magnetic architectures with tunable topology and functionalities surpassing their planar counterparts. Here, we experimentally reveal that 3D soft magnetic wireframe structures resemble compact manifolds and accommodate magnetic textures of high order vorticity determined by the Euler characteristic, $\chi$. We demonstrate that self-standing magnetic tetrapods (homeomorphic to a sphere; $\chi = +2$) support six surface topological solitons, namely four vortices and two antivortices, with a total vorticity of $+2$ equal to its Euler characteristic. Alternatively, wireframe structures with one loop (homeomorphic to a torus; $\chi = 0$) possess equal number of vortices and antivortices, which is relevant for spin-wave splitters and 3D magnonics. Subsequent introduction of $n$ holes into the wireframe geometry (homeomorphic to an $n$-torus; $\chi < 0$) enables the accommodation of a virtually unlimited number of antivortices, which suggests their usefulness for non-conventional (e.g., reservoir) computation. Furthermore, complex stray-field topologies around these objects are of interest for superconducting electronics, particle trapping and biomedical applications.

The behavior of any physical system can be described based on parameters invariant upon transformations in time and space. This includes not only general conservation laws, such as for energy or momentum, but also more specific topological properties of a manifold, which defines the shape of an object[1]. For instance, all homeomorphic compact manifolds (manifolds without boundaries which can be smoothly transformed one into another) have the same Euler characteristic, $\chi$. If objects corresponding to such manifolds have

[1]Helmholtz-Zentrum Dresden-Rossendorf e.V., Institute of Ion Beam Physics and Materials Research, Bautzner Landstr. 400, 01328 Dresden, Germany. [2]Kyiv Academic University, 03142 Kyiv, Ukraine. [3]Physikalisches Institut, Johann Wolfgang Goethe-Universität Frankfurt am Main, Max-von-Laue-Str. 1, 60438 Frankfurt am Main, Germany. [4]Helmholtz-Zentrum Berlin für Materialien und Energie, Albert-Einstein-Str. 15, 12489 Berlin, Germany. [5]Department of Physics and Astronomy, Uppsala University, Box-516, Uppsala SE-751 20, Sweden. [6]Wallenberg Initiative Materials Science for Sustainability, Uppsala University, 75121 Uppsala, Sweden. [7]University of Vienna, Faculty of Physics, Nanomagnetism and Magnonics, Superconductivity and Spintronics Laboratory, Währinger Str. 17, 1090 Vienna, Austria. [8]University of Vienna, Vienna Doctoral School in Physics, Boltzmanngasse 5, A-1090 Vienna, Austria. [9]Present address: Department of Interface Science, Fritz-Haber-Institut der Max-Planck-Gesellschaft, Faradayweg 4 - 6, 14195 Berlin, Germany. ✉e-mail: o.volkov@hzdr.de; o.pylypovskyi@hzdr.de; porrati@physik.uni-frankfurt.de; d.makarov@hzdr.de

a vector field associated with them, the Poincaré–Hopf theorem provides a link between the number of "holes" (genus) in the manifold and the total vorticity $Q^\Sigma$ of this vector field.

In the particular case of magnetism, this vector field is represented by the texture of the magnetic order parameter. The topology of a compact manifold (e.g., compactified plane $\mathbb{R}^2 \cup \{\infty\}$, sphere $\mathbb{S}^2$ or torus $\mathbb{T}^2$) uniquely determines the total vorticity including all sinks, sources and saddle points of the magnetization distribution. For infinite easy-plane magnets, exchange and magnetostatic interactions commonly assure a uniform ground state. This is a direct consequence of the constraint $\chi_{\text{plane}} = 0$, which allows for the appearance of vortices ($Q_v = +1$) and antivortices ($Q_{av} = -1$) as pairwise excitations only (Fig. 1a). This geometry is homeomorphic to a sphere with a hole, that represents infinity (Fig. 1b). This geometry transformation also involves the mapping of the magnetization field on a sphere. When considering instead a full sphere in real space ($\chi_{\text{sphere}} = +2$), the magnetic texture necessarily develops two vortices with out-of-plane cores[2], see Fig. 1c, assuring the total magnetic vorticity $Q^\Sigma = +2 \equiv \chi_{\text{sphere}}$. Even being homeomorphic to a sphere, complex 3D shapes can sustain more vortices and antivortices still keeping $Q^\Sigma = \chi_{\text{sphere}}$ as exemplified for the $N$-pod geometry in Fig. 1d, e. Although for this object $\chi = +2$, the shape supports $N$ vortices and $(N-2)$ antivortices in equilibrium, providing $(2N-2)$ topological solitons in total. 3D surfaces favor stabilization of multiple topological solitons within one object. Such systems accommodating magnetic solitons with many degrees of freedom are appealing for fundamental research in non-linear physics (e.g., strongly and weakly interacting

gases of vortices and antivortices) and applications in unconventional computational techniques such as in neuromorphic and reservoir computing[3,4]. Moreover, the concrete configuration of magnetic texture can be tuned while preserving the topology of the sample with a choice of its geometric symmetry, see Fig. 1f. For example, the plane tetrapod (angle of rotation $\alpha = 0°$) belongs to the $D_{4h}$ point symmetry group having a principal axis, and supports a Bloch line through the origin (Fig. 1g). In contrast, the tetrapod with $\alpha = 90°$ rotation of the top part belongs to the $T_d$ symmetry group with no principal axis (Fig. 1h). The latter breaks the Bloch line realizing two antivortex surface states instead of a bulk antivortex. To describe this transition it is instructive to calculate the corresponding $\mathbb{S}^1$-winding number along the closed loop,

$$\eta = \oint_{\delta S} \partial_s \phi \, ds, \tag{1}$$

where $\phi$ is the magnetic azimuthal angle, $s$ is a parameter of the $\delta S$ loop over the surface $S$ formed by the cross-section. The transition between the equilibrium bulk antivortex state ($\eta = -1$) and two surface antivortices ($\eta = 0$) occurs continuously with the change of $\alpha$, accompanied by the expansion of the homogeneously magnetized Bloch line. In particular, the characteristic size of the Bloch line changes from several exchange lengths[5] (for the case of the $D_{4h}$ symmetry) to consume the entire connection region (for the case of the $T_d$ symmetry), see Fig. 1i and Supplementary Section 1. The symmetrical branches around $\alpha = 90°$ are a consequence of the

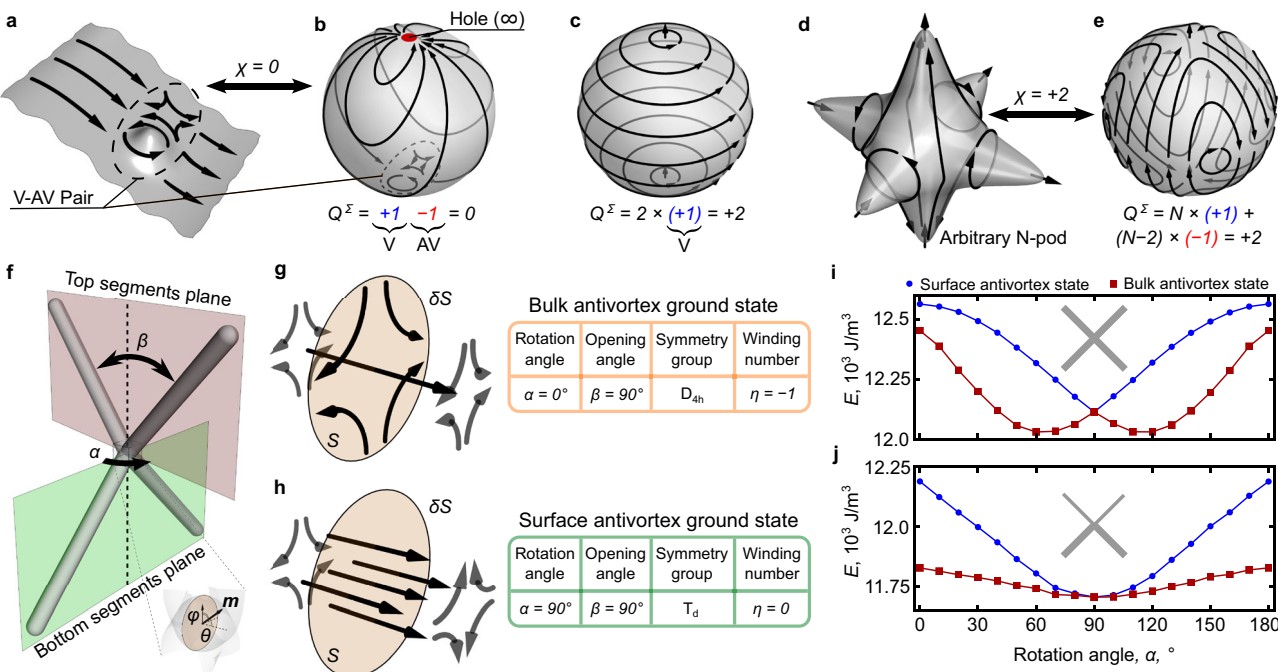

**Fig. 1 | Magnetization mapping for compact manifolds: Soft magnetic tetrapod.**
**a** Schematics of an infinite easy-plane magnetic thin film with homogeneous magnetization, which can support localized vortex (V) - antivortex (AV) pairs. The vorticity of the vortex (antivortex) is +1 (−1). **b** Corresponding mapping of the texture shown in a on the unit sphere with a hole. The Euler characteristic of this geometry is $\chi = 0$. Black arrows depict the magnetization distribution on a surface. **c** Two vortices are formed in equilibrium at the poles of a complete sphere in line with the hairy ball theorem. The Euler characteristic of a sphere is $\chi_{\text{sphere}} = +2$. **d** The stability of a complex magnetic texture with $N$ vortices and $(N-2)$ antivortices in an $N$-pod and **e** its mapping onto a unit sphere. The total vorticity is $Q^\Sigma = +2 \equiv \chi_{\text{sphere}}$ as the $N$-pod is homeomorphic to a sphere. **f** Tetrapod geometry constructed of four line segments with a length $L = 1.6\,\mu m$ and circular cross-section with radius $r = 58$ nm. The opening angle between the two top and two

bottom line segment pairs is $\beta = 90°$. Each pair of line segments forms a plane. These planes are rotated by the rotation angle $\alpha$ with respect to each other. **g, h** Schematic illustrations of the bulk and surface antivortex states, respectively, that form in the center interconnection region of tetrapods with $\alpha = 0°$ and $\alpha = 90°$, respectively. The planes $S$ represent the target space for the calculation of the winding number of the corresponding magnetization distributions depicted by the black arrows. Dependencies of the total energy of the two states, which define bulk (red squares) and surface (blue disk) antivortices on the rotation angle $\alpha$ for the (**i**) symmetric (all line segments have the radius $r = 58$ nm) and (**j**) asymmetric tetrapods (bottom line segments have $r = 58$ nm, while top ones have $r = 40$ nm). At $\alpha = 90°$ both states coincide up to a symmetry transformation. For the asymmetric tetrapod, there is a wider area by $\alpha$, where the energies of these states are almost equal.

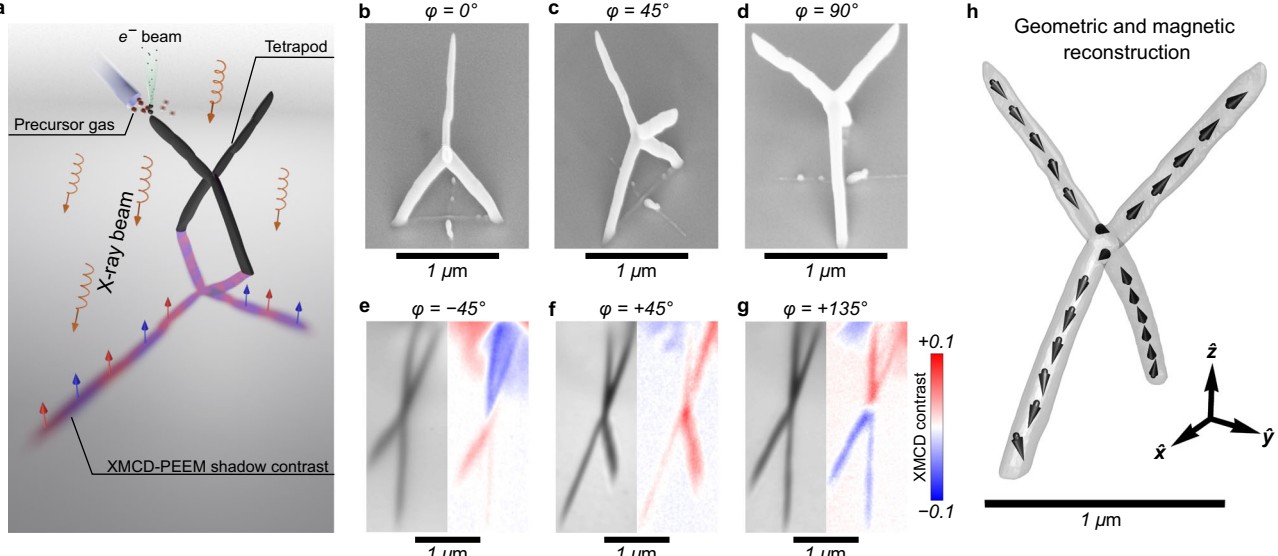

**Fig. 2 | Experimental realization and investigation of a soft magnetic tetrapod. a** Schematics of a FEBID fabricated $Co_3Fe$ tetrapod and magnetic imaging in the shadow by mean of XMCD-PEEM. Orange helical arrows indicate the direction of the synchrotron X-ray beam. Blue and red arrows indicate intensities of emitted photoelectrons from the surface. **b**–**d** Scanning electron microscopy images of the tetrapod structure taken at different azimuthal angles: $\phi = 0°$, $\phi = 45°$ and $\phi = 90°$.

**e**–**g** PEEM image and the corresponding XMCD-PEEM shadow contrast of the tetrapod imaged at different azimuthal angles with respect to the incident X-ray beam: $\phi = -45°$, $\phi = +45°$ and $\phi = +135°$, respectively. **h** Schematics of the magnetization distribution in the tetrapod derived from the set of XMCD-PEEM images. Further experimental tetrapod geometries and their XMCD-PEEM characterization are presented in Supplementary Sections 4 and 5.

identical shapes and at the same time reflect the duality regarding the location of the antivortices. Only for $\alpha = 90°$, states with the surface and bulk antivortices become symmetry-connected and, accordingly, equal in energy, while for all other $\alpha$ the bulk antivortex has slightly lower energy. An increase of the angular range where both these states have almost the same energies ($80° \lesssim \alpha \lesssim 100°$) can be achieved if the top and bottom tetrapod parts are asymmetric, e.g., the diameter of top and bottom line segments is different, see Fig. 1j. This type of asymmetric geometry is of advantage for experimental explorations aiming to realize surface antivortex textures. This enables lower magnetization gradients in the connection regions of the tetrapod line segments and favors the internal part of the connection region to be homogeneously magnetized.

Curvilinear and 3D self-standing architectures are explored for the realization of chiral non-collinear magnetic textures[6], 3D magnetic interconnects[7], topological magnetic field nanotextures[8] and 3D racetrack memory devices[9]. These experimental activities benefit from the firm theoretical framework of curvilinear micromagnetism[10,11], which offers the possibility to tailor fundamental magnetic responses relying on the geometry of the object. Geometrically curved nanostructures are fabricated by means of advanced nanofabrication techniques which include two-photon lithography[12,13], charged aerojet nanoprinting[14] and focused electron-beam-induced deposition (FEBID)[15–18]. In particular, these methods allow creation of magnetic objects with shapes corresponding to compact manifolds. For instance, an $N$-pod (e.g., see Fig. 1d) resembles a voluminous object surrounded by a surface without boundary and can be fabricated, e.g., by FEBID[19], as a wireframe structure based on appropriately connected soft magnetic nanowire segments.

Here, we propose and experimentally validate a methodology to realize complex magnetization textures with large total vorticity $|Q^{\Sigma}|$ in equilibrium. Using FEBID, $Co_3Fe$ free-standing tetrapods, pyramid- and cube-wireframes are fabricated and imaged by means of X-ray magnetic circular dichroism photoelectron emission microscopy (XMCD-PEEM), see Fig. 2a. By combining experimental data and theoretical analysis, we show that the tetrapod supports six solitons (four vortices and two antivortices) in equilibrium, whose location is controlled by

the geometry of the object. We demonstrate that the tetrapod's geometrical structure can be tailored in such a way as to realize a surface antivortex state, which is stabilized when the uniformly magnetized bulk region between segments breaks the Bloch line connecting two adjacent antivortices. Non-collinear magnetization textures with six solitons on the tetrapod result in the formation of complex topological magnetic stray field patterns surrounding the object. These topological magnetic field nanotextures are analyzed as well and their application potential for nanoscale robotics and superconducting electronics is outlined. Furthermore, we explore theoretically and experimentally distinct wireframe structures possessing different Euler characteristics ranging from positive (+2, objects homeomorphic to a sphere) through 0 (objects homeomorphic to a torus) to negative values (torus with a certain number of holes, $n$-torus). For geometries featuring $\chi = 0$, a balance between the number of vortices and antivortices may be of interest for constructing 3D spin-wave splitters[20]. In contrast, objects with $\chi < 0$ represented by pyramids and cubes provide the unique possibility to stabilize a virtually unlimited number of magnetic solitons with prevailing number of antivortices. To this end, we show that a pyramid-shaped wireframe can support six antivortices without vortices, while a cube-shaped structure allows to stabilize eight surface antivortices without vortices. Because of the topological stability of solitons of similar kind, even upon interaction they are stable against annihilation, which suggests their relevance for applications in low-power computation schemes utilizing paradigms of reservoir[3,4] and probabilistic[21,22] computing.

## Results

### Topologically non-trivial surface states in tetrapods

According to Fig. 1i, j the surface antivortex state can have similar total energy to the bulk antivortex. To find appropriate geometric parameters allowing stabilization of surface states, we perform extensive micromagnetic simulations varying length and radius of the tetrapod line segments, and opening angle, see Supplementary Section 2. In this respect, tetrapod geometries with $\alpha = 90°$, $\beta > 60°$, $L > 1\,\mu m$ and $40 \leq r \leq 70$ nm represent the most suitable conditions for the formation of topologically non-trivial surface states in experiment.

Moreover, these geometric parameters are well achievable with the FEBID technique. Hence, FEBID is used for the fabrication of a self-standing magnetic tetrapod, see Fig. 2a and Supplementary Section 3. The fabrication is done on a gold-coated Si wafer by means of electron beam induced dissociation of the precursor $HCo_3Fe(CO)_{12}$ into $Co_3Fe$ alloy and carbon leftovers, that partially remain in the fabricated structure[17], see Fig. 2a. The $Co_3Fe$ alloy is ferromagnetic at room temperature, revealing soft magnetic properties due to the nano-crystalline structure and carbon content[23]. The resulting tetrapod geometry is shown in Fig. 2b–d. With a total height of 1.9 μm, it is constructed from four 1.3 μm long and 110 nm thick nanowire segments with opening angles of 90° and 72° between the top and bottom branches, respectively. The top part of the tetrapod has a 90° azimuthal rotation with respect to the bottom part. In our work we benefit from prior studies of different magnetic wireframes accommodating complex magnetization textures[17,19,24–27].

The magnetic states are visualized at remanence by means of XMCD-PEEM through shadow contrast imaging at the $L_3$ absorption edge of Co after an in-plane AC demagnetization procedure with a maximum magnetic field of 100 mT. In the case of a planar sample, the XMCD contrast from the surface reveals magnetic vectors being parallel (red), antiparallel (blue) and perpendicular (white) to the direction of the X-ray beam. In the case of 3D magnetic samples, the resulting shadow XMCD contrast is inverted due to the helicity-dependent absorption, i.e., parallel magnetization alignment in the 3D object with respect to the beam corresponds to the blue contrast, while the antiparallel one is colored in red[28].

Due to the high symmetry of the tetrapod, its magnetic state can be reconstructed by performing XMCD-PEEM imaging from three different azimuthal angles $\phi$ being $-45°$, $+45°$ and $+135°$ with respect to the bottom tetrapod part positioned along $\phi = 0°$, see Fig. 2e–g. Due to the branching of the magnetic segments of the tetrapod, such diagonal scans allow not only to obtain the sample geometry, but also to reconstruct the magnetization distributions in all segments of the tetrapod (Fig. 2h). Namely, each segment of the tetrapod is observed to have its magnetization distribution being directed primarily along its axis. Hence, the total magnetization pattern of the tetrapod resembles an antivortex state in the central region.

For accurate state reconstruction, finite element micromagnetic simulations are performed for the experimentally obtained tetrapod geometry. The resulting magnetic state is shown in Fig. 3a. It coincides with the reconstructed magnetic distribution obtained from the XMCD-PEEM experiments (Fig. 2h and Supplementary Section 5). A detailed analysis of the magnetic streamlines inside the simulated tetrapod geometry (Fig. 3b) reveals the presence of a mainly uniform magnetization distribution inside the tetrapod volume and in total six non-trivial surface solitons: two antivortices at the connection area (Fig. 3c) and four surface vortices pinned at the ends of the tetrapod segments (Fig. 3d). To determine the location of these solitons, we calculate the flux density of the topological charge, which is suitable for complex 3D magnetic objects[29]. The observed magnetization state is determined by geometric connections of the four wire segments. For the given sample dimensions, it is possible to realize a uniform magnetization field in each of the segments, which terminates in a vortex. The connection region of the four wire segments breaks the spatial symmetry and leads to the appearance of divergent magnetization fluxes, manifested in the formation of antivortices.

The distinct feature of the observed vortices and antivortices is that they represent surface states, in contrast to the commonly observed volume state textures in nanodots. In the experimental geometry, two antivortices do not share a common Bloch line, which joins their centers. Instead, their individual Bloch lines disappear towards the depth within several dozens of nanometers (Supplementary Section 6). This turns the central part of the tetrapod to be almost homogeneously magnetized along the direction connecting the two

antivortices (Fig. 3c). These surface states are observed in tetrapod geometries constructed with four wire segments, which are pairwise accommodated in orthogonal planes, i.e., the experimental tetrapod geometry. In contrast, simulations of tetrapods whose wire segment pairs lie in the same plane or in planes, which are rotated by $\alpha = 45°$, reveal the presence of a Bloch line shared by both antivortices at the surface (Supplementary Section 2). The latter resembles the bulk-like antivortex texture observed in planar nanodots[30,31]. For the diameter of wire segments forming the tetrapod in our experiment, the magnetization curls, forming the vortex when approaching the end of each segment (Fig. 3d). Similarly to antivortices, vortex Bloch lines are well-defined only near the end of the segment. We note that the recently reported site-selective vapour deposition using FEBID[32] allows the realization of magnetic nanoshells of $Co_3Fe$ decorating curvilinear wires made of PtC. Such surface modification may allow one to obtain a easy-normal magnetic anisotropy and intrinsic Dzyaloshinskii-Moriya interaction (DMI), which could enable the investigation of curvature-induced skyrmions[33,34] in curvilinear FEBID-fabricated nanoshells.

To demonstrate that the total vorticity over the tetrapod surface is equal to the Euler characteristic of the geometry being $\chi = +2$, we perform a homeomorphic transformation of the tetrapod into a unit sphere positioned in a tetrapod centroid keeping correspondence between the magnetization direction and vector field on a sphere, see Fig. 3e. Figure 3f shows the angular projection of the sphere in azimuthal and polar coordinates. The total vorticity of the texture is $Q^\Sigma = 4 \times (+1) + 2 \times (-1) = +2$ in line with the Euler characteristic for a sphere without holes, $\chi_{sphere} = +2$. Moreover, the external magnetic field applied to the tetrapod geometry does not change the total vorticity (Supplementary Section 6).

## Discussion
### Design of high-order vorticity states

As the total vorticity of a wireframe geometry is determined by its Euler characteristic, this opens the possibility to design complex magnetization patterns through the fabrication of magnetic geometries with specific topological properties. Namely, the generalized $N$-pod geometry being homeomorphic to a sphere assures the total vorticity to be equal to $Q^\Sigma = +2$ as for the tetrapod ($N = 4$) with four vortices and two antivortices. The change of the number of pods in the $N$-pod only adds or removes a vortex-antivortex pair but does not affect the total vorticity (Supplementary Section 7). The pentapod wireframe geometry ($N = 5$) with five line segments obtains in total five surface vortices and three antivortices. In particular, the vortices are necessarily located at the ends of nanowires forming the pentapod, while three antivortices can take positions in the central part according to the symmetry of the geometry (Supplementary Section 7C). In the particular case of the pentapod geometry presented in Fig. 4a–c, one antivortex is at the side of the pentapod and two antivortices are at the bottom of the junction area. According to the magnetization streamlines, they are separated by locally homogeneously magnetized volumes (Supplementary Section 7C). In this sense, there are no well-defined Bloch lines associated with pairs of vortices and antivortices because each topologically non-trivial texture has a smooth transition to the uniformly magnetized region within the segment. The formation of the observed surface antivortices is forced by the necessity to connect the uniformly magnetized volumes as smoothly as possible to minimize the exchange energy. We note that for the case of three antivortices, symmetry prohibits their connection by Bloch lines. Therefore, at least one of the antivortices will always represent the surface state for any configuration of the pentapod.

Purely geometric symmetries do not allow stabilization of surface topological solitons with vorticity higher than ±1, which are penalized by the exchange energy. For example, a star-like geometry with $C_6$ symmetry could be expected to favor the formation of an antivortex with $Q = -2$ on the top and bottom sides. However, this texture decays

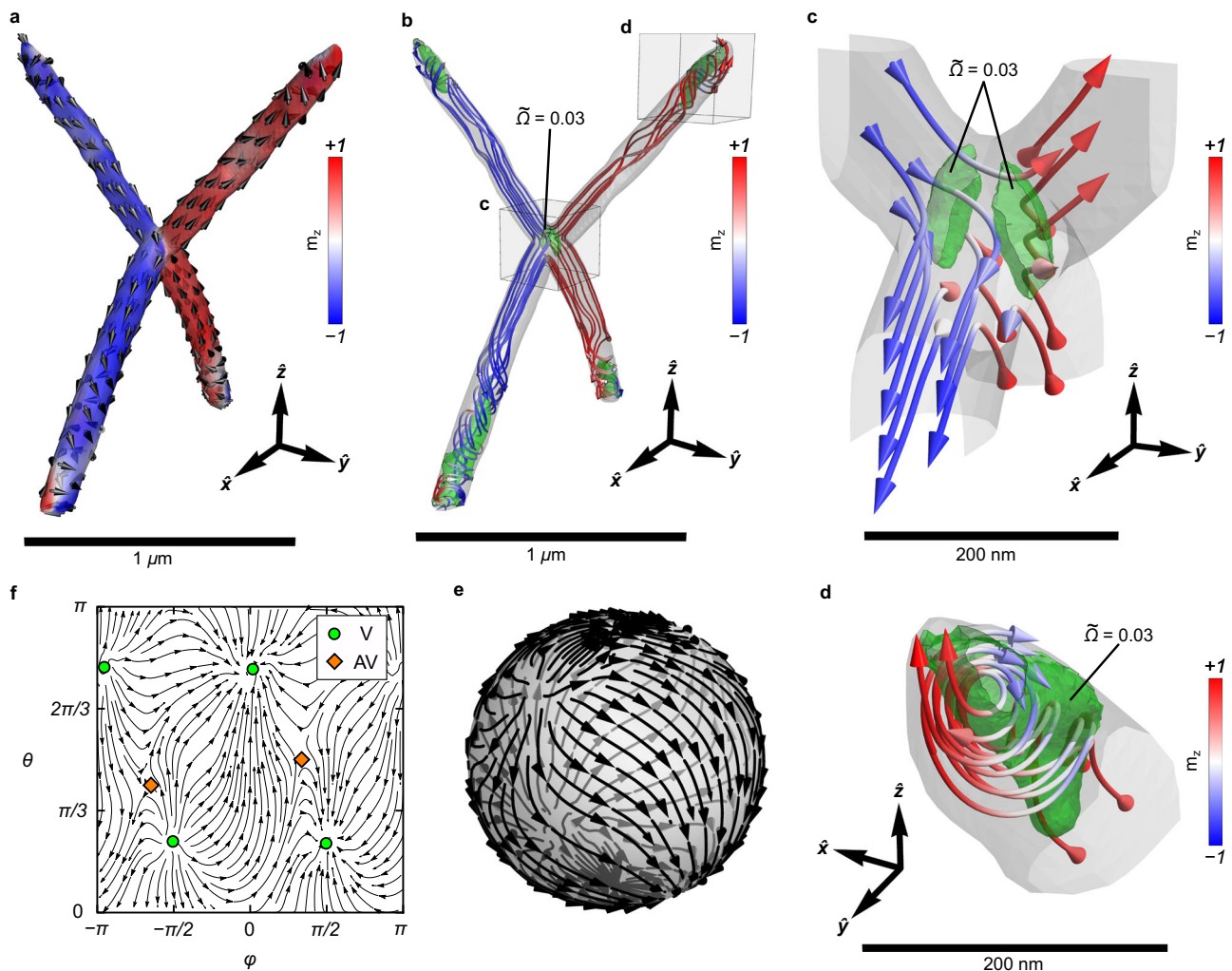

**Fig. 3 | Topological magnetization textures with high-order vorticity in the tetrapod geometry. a** Magnetic state of the tetrapod simulated using full-scale micromagnetics. The geometry in simulations is based on the reconstruction of the shape of the object studied experimentally. Arrows indicate magnetization direction. The color scheme shows the distribution of the $m_z$ component in the sample. **b** Streamlines of the internal magnetization distribution in the tetrapod geometry. Green regions show isosurfaces of the topological charge density $\widetilde{\Omega} = 0.03$. These regions correspond to the location of non-trivial magnetic states, namely **c** surface antivortices and **d** vortices. **c** Two surface antivortices in the central region of the tetrapod, which appear due to mutually perpendicular regions with uniform magnetization in the top and bottom wire segments. The region between these

antivortices restores a homogenous magnetization distribution. For this tetrapod geometry, surface antivortices are energetically favorable compared to a bulk antivortex state (two antivortices connected with a Bloch line). **d** The surface vortex at the end of the top wire segment. The isosurface of the topological charge density is indicated by the green region. **e** The surface distribution of magnetization of the tetrapod mapped onto a unit sphere and **f** corresponding angular projection of the sphere in azimuthal and polar coordinates. Green circles indicate the position of vortices and orange diamonds show the position of antivortices. Further details on surface and bulk solitons in tetrapod geometries are summarized in Methods and Supplementary Section 2.

into a pair of two closely spaced antivortices with $Q = -1$ on each side, even for geometries with short nanowires with a diameter of about the exchange length (Supplementary Section 7A). Higher-order topological magnetic solitons may be achieved in materials with strong enough next-to-nearest neighbor exchange introducing additional micromagnetic energy terms[35,36].

Additive nanofabrication could also be utilized for the construction of magnetic wireframes of more complex topology, that are homotopic to the so-called $n$-torus ($\mathbb{T}^n$). This wireframe structure has $n$ holes and its corresponding Euler characteristic is $\chi_{n\text{-torus}} = 2(1-n)$. For instance, upon FEBID fabrication of magnetic tetrapod geometries directly on the substrate surface, an additional co-deposited magnetic layer appears in the bottom part of the geometry due to the electron beam scattering[17]. Such linkage introduces a change in the geometric topology being homeomorphic to a regular torus with one hole and $\chi_{\text{torus}} = 0$. As a result, this object should support formation of an equal

number of surface solitons of positive and negative vorticity. An exemplary pentapod with a loop is shown in Fig. 4d–f. The magnetic state in a soft magnetic torus acquires three vortices at the segment ends and three antivortices in the central part ($Q^\Sigma = 0$). This discussion is in line with the literature on torus-shaped ferromagnets[37]. Two additional vortices observed in Fig. 4a–c for the parent pentapod structure are now eliminated by the presence of the horizontal connecting segments. These wireframe structures accommodating a loop and homeomorphic to a torus may be of interest for constructing various 3D spin splitters in the frame of higher-order vortex-antivortex nanostructures that could channel spin-waves along domain walls[20].

Further design of higher-order vorticity is possible with $n$-torus with $n > 1$. This is illustrated by a wireframe pyramid, which is equivalent to a 4-torus (Fig. 4g–i). Being characterized by the Euler characteristic $\chi_{4\text{-torus}} = -6$, this geometry supports more antivortices than vortices. For the particular case shown in Fig. 4g (see also

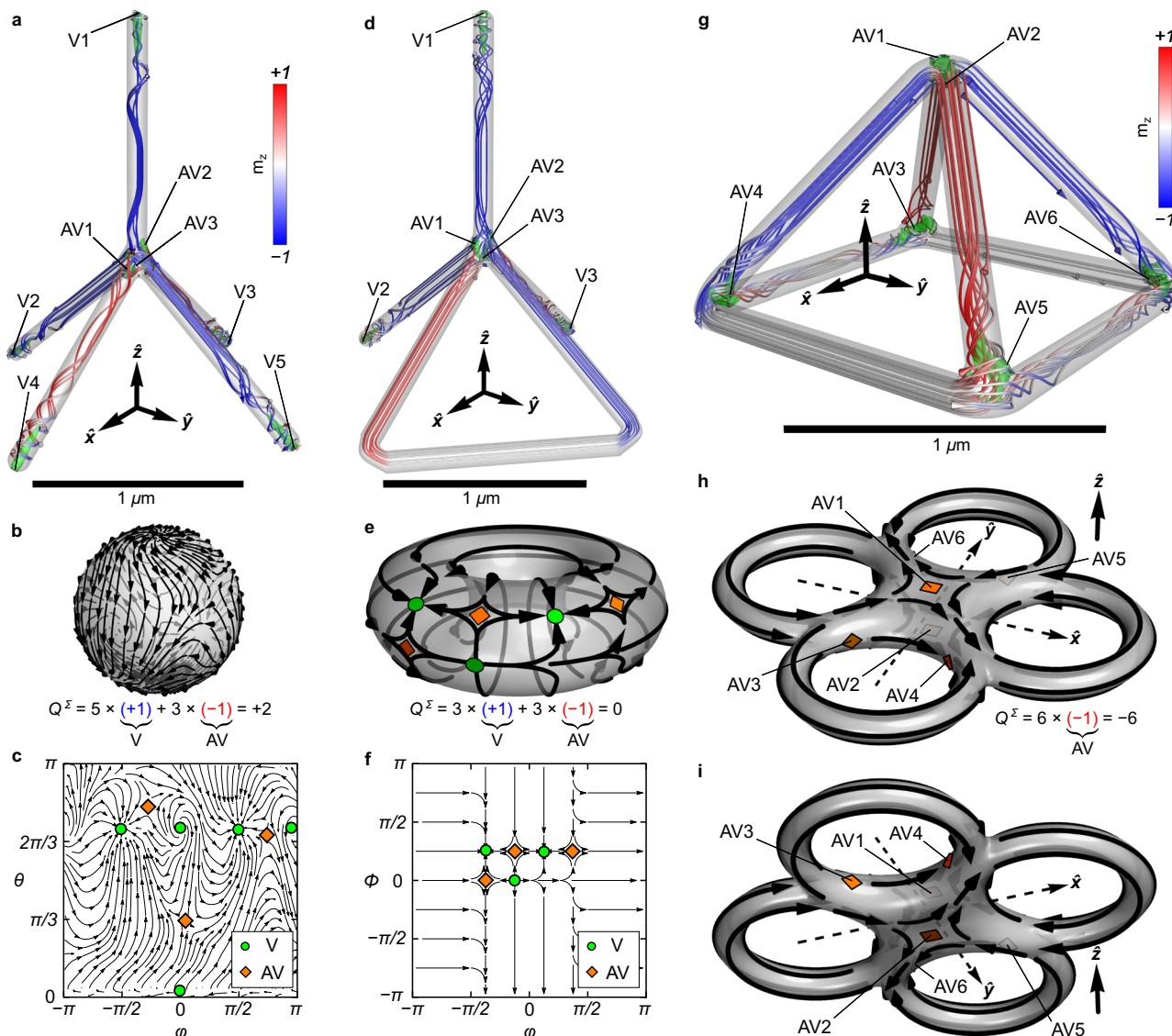

**Fig. 4 | Topological magnetization textures with high-order vorticity in wire-frames of different Euler characteristic. a** Magnetization state of a symmetric pentapod represented by the corresponding streamlines. Isosurfaces of the topological charge density distribution show the position of non-trivial magnetic surface textures providing in total five vortices (V) and three antivortices (AV). In this and other panels, the position of magnetic solitons is indicated with streamlines and green regions show isosurfaces of non-zero topological charge density. The total vorticity of the 8 solitons is $Q^\Sigma = +2$. This matches the Euler characteristic of this object $\chi = +2$, which is homeomorphic to a sphere. **b** Mapping of the magnetization distribution of the pentapod surface onto a unit sphere and (**c**) corresponding angular projection of the sphere in azimuthal and polar coordinates. Green circles indicate the position of vortices and orange diamonds show the position of antivortices. **d** Magnetization state of a pentapod with a loop represented by the corresponding streamlines. This geometry has Euler characteristic

$\chi = 0$. This object is homeomorphic to a torus. The total vorticity of this object should be $Q^\Sigma = 0$. For the shown geometry, this is reflected in the presence of three vortices and three antivortices. **e** Schematic mapping of the magnetization distribution of the surface of the pentapod with a loop onto a torus and (**f**) corresponding schematics of the projection to the toroidal azimuthal and polar angles $\phi$ and $\Phi$, respectively. **g** Magnetization state of a wireframe pyramid formed by 8 segments. This object is homeomorphic to a 4-torus, i.e., torus with 4 holes. The Euler characteristic of a 4-torus $\chi = -6$. For the shown geometry, the equilibrium state consists of 6 antivortices, namely four surface and two bulk antivortices, resulting in a total vorticity of $Q^\Sigma = -6$. Schematics of the mapping of the magnetization distribution of the surface of a pyramid on a 4-torus: **h** top view under an angle and (**i**) bottom view under an angle. Further examples of magnetic wireframes are discussed in Supplementary Section 7.

Supplementary Section 7D), it contains only six antivortices in equilibrium. The shown magnetization configuration has high symmetry and contains two antivortices at the top apex, which are connected by a Bloch line (bulk antivortex state). Four antivortices are located in each of the bottom corners of the pyramid and are surface states because of the absence of Bloch lines connecting them in pairs. Similarly, the construction of wireframe geometries with a higher number of holes results in an increasing number of antivortex states. Namely, a cube wireframe geometry being equivalent to a 5-torus with $\chi_{5-torus} = -8$ enables the formation of eight surface antivortices in the

vertices, see Supplementary Note 7E. Thus, introducing additional holes into the wireframe geometry leads to the appearance of more antivortices at the wireframe vertices increasing the total vorticity of the system. In particular, the wireframe buckyball geometry constructed from 90 nanowires connected at 60 vertices[27] should have the Euler characteristic $\chi_{buckyball} = -60$ and accordingly contain at least 60 surface antivortices with $Q = -1$ each.

These theoretical predictions can be readily confirmed experimentally by characterizing more complex structures prepared by FEBID including magnetic semi-pyramid, diamond and cube

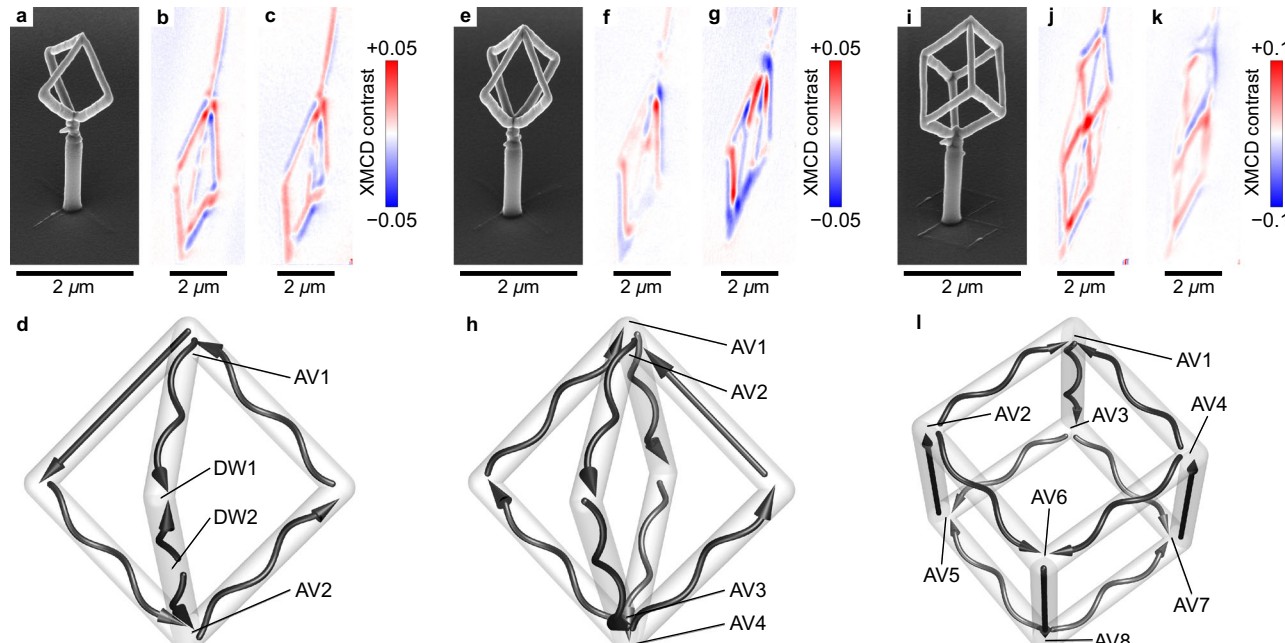

**Fig. 5 | Experimental investigation of magnetic wireframe geometries with high-order vorticity. a** Side-view scanning electron microscopy (SEM) image of a magnetic semi-pyramid and corresponding XMCD-PEEM shadow contrasts taken at different azimuthal angles of (**b**) $\phi = 0°$ and (**c**) $\phi = -25°$. Homogeneously magnetized line segments provide a monochrome shadow contrast being blue for the magnetization pointing parallel to the beam and red when it is antiparallel. These segments have diameter of around 130 nm and obtain homogeneous magnetic state. The formation of dipolar contrast in a line segments indicate on the appearance of vortex-type magnetization rotation, which is typical for the nanowires with a diameter of around 180 nm used for the wireframe formation. **d** Schematics of the magnetization distribution in the semi-pyramid derived from the set of XMCD-PEEM images. Black arrows indicate the magnetization direction in each line segment of the wireframe geometry. This object is homeomorphic to a 2-torus. Hence, it accommodates 2 surface antivortices. **e** Side-view SEM image of a magnetic diamond wireframe geometry and corresponding XMCD-PEEM shadow

contrasts taken at different azimuthal angles of (**f**) $\phi = 0°$ and (**g**) $\phi = -25°$. **h** Schematics of the magnetization distribution in the diamond wireframe derived from the set of XMCD-PEEM images. This object contains two bulk antivortices at the top and bottom vertices of the diamond. These two bulk antivortices correspond to 4 surface antivortices, which is in line with the Euler characteristic of this geometry, $\chi_{3-torus} = -4$. **i** Side-view SEM image of a magnetic cube wireframe geometry and corresponding XMCD-PEEM shadow contrasts taken at different azimuthal angles of (**j**) $\phi = 0°$ and (**k**) $\phi = -25°$. **l** Schematics of the magnetization distribution in the cube wireframe derived from the set of XMCD-PEEM images. This object accommodates 8 surface antivortices, which agrees with its Euler characteristics, $\chi_{5-torus} = -8$. Each of the magnetic wireframes shown in this figure is based on $Co_3Fe$ alloy written by FEBID. To enable full magnetic characterization using XMCD-PEEM, the objects are fabricated on a non-magnetic PtC pillar, which is also prepared by FEBID.

wireframes, see Fig. 5 and Supplementary Section 4. Each magnetic geometry is grown on a non-magnetic PtC pillar, which elevates it above the substrate and provides access to the magnetization configuration of the entire object through the XMCD-PEEM imaging. These non-magnetic PtC pillars are covered with a layer of $Co_3Fe$ due to its co-deposition upon FEBID preparation of the magnetic wireframes. Still, the magnetic coating on the pillar does not change the resulting Euler characteristic of the magnetic geometry. Indeed, this additional coating does not lead to the formation of holes in the system. Thus, its influence on the overall topological picture of the wireframe geometry is negligible. We note that all structures are imaged in the as-grown state without exposing the samples to magnetic fields before imaging. For the case of semi-pyramid (homeomorphic to 2-torus), we observe the formation of two surface antivortices (Fig. 5d) in line with the Euler characteristic of this geometry, $\chi_{2-torus} = -2$. Introducing additional hole to the magnetic object through the construction of a diamond-shaped wireframe structure (Fig. 5e), we obtain a geometry characterized by $\chi_{3-torus} = -4$ and accommodating two bulk antivortices with the corresponding 4 surface antivortices. Reconstruction of the magnetic state of a cube wireframe (homeomorphic to a 5-torus with $\chi_{5-torus} = -8$; Fig. 5i) reveals the presence of eight surface antivortices located at cube vertices (Fig. 5l; see also Supplementary Note 7E).

Magnetic wireframes hosting a large number of solitons could be considered as a platform for the realization of physical magnetic reservoirs for neuromorphic computing as they fulfill requirements on

(i) interconnection complexity, (ii) reproducibility of reservoir states and (iii) non-linear interaction of system components[38]. Namely, FEBID offers possibilities to realize complex wireframe structures as it was demonstrated, e.g., for the buckyball geometry[39] supporting at least 60 antivortices. The reproducibility of the state is assured by the link between the total vorticity of the surface magnetization and the Euler characteristic of the geometry. The interaction strength can be tuned by selection of the length and diameter of the wireframe's segments, which will enrich their non-linear spatio-temporal dynamics. The formation of magnetic states with only antivortex textures may be beneficial as it prevents texture annihilation, which ensures the fading memory criteria of the physical reservoir. The direct integration of nanofabricated complex 3D wireframes into standard 2D lithographically created systems[40] with coplanar or Ω-shaped antennas or detectors should allow extending unconventional computing into 3D, offering additional functionalities such as a higher degree of interconnectivity.

## Topology of magnetic field nanotextures

Magnetic wireframes provide a platform to design topological magnetic field nanotextures, as was introduced for interacting magnetic double helices[8]. Benefiting from the complexity of their magnetic states possessing higher-order vorticity, soft magnetic wireframes enable vast capabilities in shaping of the magnetic near field. We envision that a strategy to tailor field nanotextures on a specific

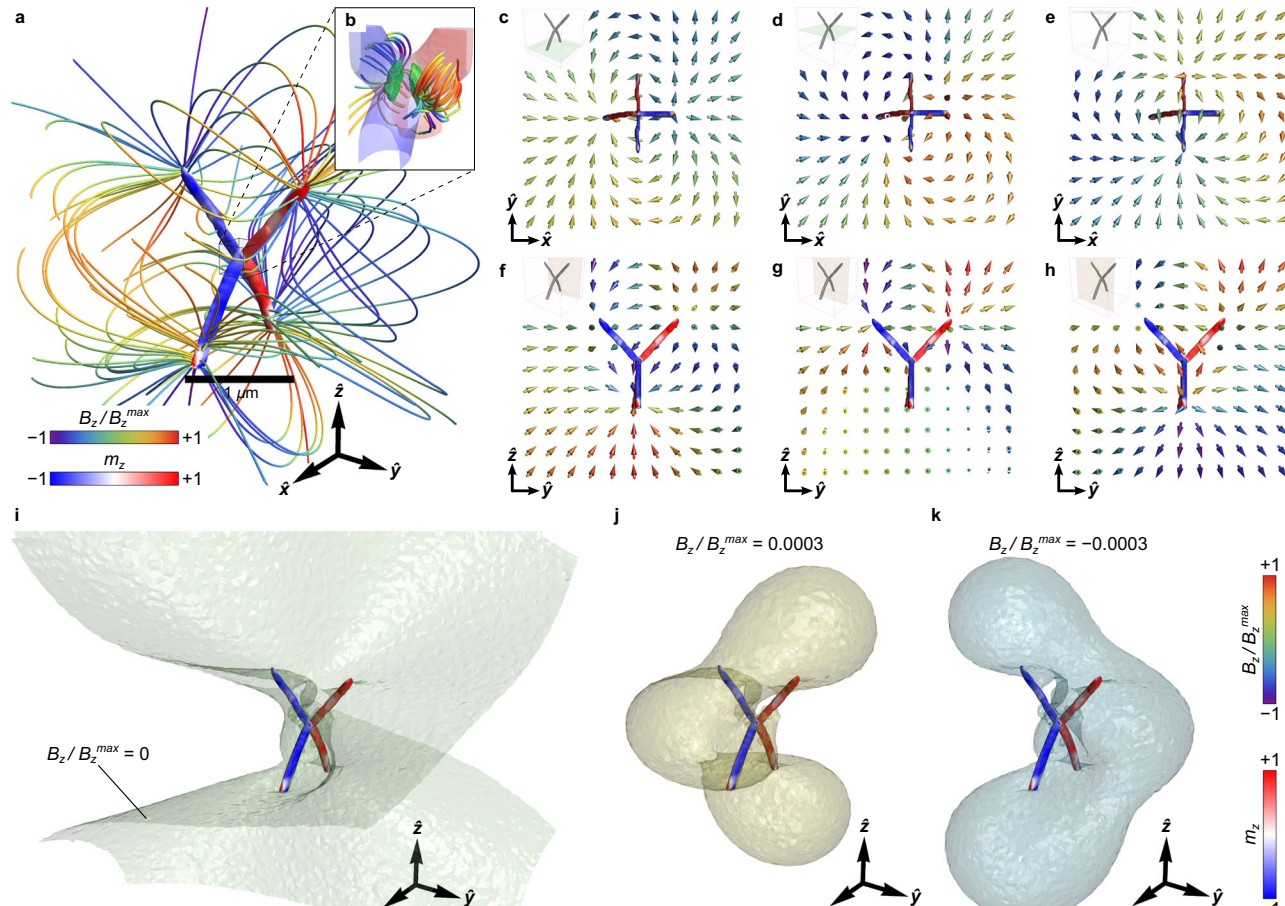

**Fig. 6 | Near-field magnetic field textures around the magnetic tetrapod.**
**a** Streamlines of the magnetic field that have been obtained from the reconstructed magnetic state of the FEBID-grown tetrapod (Fig. 2h). The red-white-blue contrast depicts the magnetization distribution over the tetrapod surface, while the rainbow contrast represents the $B_z$ component of the stray-field distribution. **b** Streamlines of the magnetic near-field originating from two surface antivortices in the central area of the tetrapod. Green regions show isosurfaces of the topological charge

density $\widetilde{\Omega} = 0.03$. Families of **c–e** vertical and **f–h** horizontal cross-sections of the calculated distribution of the magnetic field texture. Color of individual arrows depicts the direction of the $B_z$ component. Isosurfaces of $B_z/B_z^{max}$ are bent due to the geometrical frustration of the experimental tetrapod geometry. The isosurfaces are shown for (**i**)$B_z/B_z^{max} = 0$, (**j**)$B_z/B_z^{max} = 3 \times 10^{-4}$ and (**k**)$B_z/B_z^{max} = -3 \times 10^{-4}$. Further details on topological stray field nanotextures are summarized in Supplementary Section 8.

topology is via proper geometry modifications that preserve the topological invariants, i.e., geometric transformations that are homeomorphic such as twists (Supplementary Section 8A). Such twists introduce complex transitions between the topology of the isosurfaces of the magnetic field with designable spatial field gradients, and tunable orientability of the field. For instance, the experimental tetrapod geometry favors field lines that consist of multiple loops that indicate a well defined field closure (Fig. 6a–h). These complex field lines reveal a topologically non-trivial three-dimensional stray field textures featuring coupling between the opposite segments of the tetrapod. Moreover, the distribution of the magnetic near-field around the central connection region of the tetrapod reveals a highly divergent field profile originating from the surface antivortices in that area, see Fig. 6b. Namely, each surface texture acts as a source of a strong magnetic near field, which bends into loops when moving away from the central area. These high-gradient magnetic fields may be potentially utilized for pinning and guiding of ultracold atoms by means of non-linear magnetic fields[41].

The analysis of cross-sections of the magnetic field texture shown in Fig. 6c–e reveals a 90° twist of the field induced by the sample shape (Supplementary Section 8B). The twist is visualized in Fig. 6h showing the isosurface $B_z/B_z^{max} = 0$. The middle cross-section (Fig. 6d) suggests a complex stray field texture emerging from the surface antivortices (Fig. 3b). An overview of the horizontal cross-sections in Fig. 6f–h

indicates a symmetry break, which is different from a conventional dipolar field profile. The closed isosurfaces $B_z/B_z^{max} = \pm 3 \times 10^{-4}$ around the tetrapod in Fig. 6j, k reveal two disconnected regions that are complementary under rotation by 90°. Further analysis of symmetries of $B_z$-isosurfaces for different tetrapod geometries (Supplementary Section 8) confirms that the symmetry break merges three disconnected regions into a single one (genus 1), providing a broader angular orientability around the vertical direction ($\hat{z}$-axis). The non-trivial stray field topology still preserves some transfer of symmetry from the shape along the vertical direction with respect to the isosurface in Fig. 6i.

We note that tetrapods as well as individual nanowires are homeomorphic to a sphere. In particular, pairs of surface vortices were discussed theoretically and experimentally for nanowire geometries, see e.g.,[42–44]. Individual nanowires were already explored to design stray field patterns relying on local modulations in the wire diameter and material composition[45,46]. Here we demonstrate that symmetry breaks by means of twists in magnetic wireframes allows to design the topology of three-dimensional stray field nanotextures (Supplementary Section 8).

In summary, we demonstrate the design of high-order vorticity textures in magnetic wireframe structures, which can be fabricated by additive nanofabrication methods. These objects can support numerous magnetic solitons (vortices and antivortices) in the ground state.

The number and the type of prevailing solitons (vortex vs. antivortex) is determined by the topology of the wireframe (Fig. 4). In particular, we discuss geometries homeomorphic to an $n$-torus including pyramids and cubes, which can accommodate only antivortices. Objects with a large number of solitons of the same type, which are robust against annihilation, might find application in unconventional computing schemes like reservoir computing.

The topological stability of magnetization textures in wireframe structures with high-order vorticity assures stability of magnetic stray field patterns. This suggests considering diverse application prospects of magnetic wireframes. The design of 3D magnetic field nanotextures (Fig. 6 and Supplementary Section 8A), which are stable under externally applied magnetic fields, renders geometries with higher order vorticity useful for biomedical applications. Indeed, tetrapod structures can provide strong gradients of the magnetic near field, which renders these objects relevant as components of smart micromachines for their navigation and localization in microsurgery and drug delivery[47,48]. Furthermore, complex magnetic stray field patterns enable the trapping of magnetically-functionalized objects in biomedical screening assays but also magnetic particles in ultracold environments[41,49].

3D wireframe magnetic objects offer the possibility to design appropriate magnetic stray field templates for superconducting electronics, e.g., for pinning of superconducting vortices aiming to control the electrical resistance of superconductors[50]. In particular, placement of magnetic nanoarchitectures close to the edge of a superconductor film can find use in probing the stray fields via Abrikosov vortices. The sensitivity of the electric voltage response of superconductors to the suppression of superconductivity along their edges[51] should allow for tuning the vortex trajectories from vortex chains[52] to vortex jets[53] and exploring topological transitions between them[54,55]. The integration of 3D topological objects in hierarchical systems with geometric frustration has potential to impact the area of "magnetricity"[56], which is concerned with the steering of emerging magnetic monopoles in artificial spin-ice systems[57,58].

## Methods

### Sample preparation

Tetrapods, pyramids, diamonds and cubes discussed in the main text and Supplementary Section 3, 4 were fabricated by FEBID inside a dual-beam SEM/FIB microscope (FEI, Nova NanoLab 600) equipped with a Schottky electron emitter. In FEBID, molecules of a precursor gas injected in the microscope and adsorbed on the surface of a substrate dissociate by interaction with the electron beam. The solid part of the dissociation product forms the sample during the electron beam movement[59]. A wide range of available material compositions enable the application potential of FEBID in the formation of 3D geometries for nanomechanics[60], plasmonics[61], magnonics[62], superconductivity[63] and nanomagnetism[8,18,64]. For the present study, the $HCo_3Fe(CO)_{12}$ precursor gas is used[23]. The precursor is delivered by a standard gas injection system (GIS) via a capillary with inner diameter of about 0.5 mm. The precursor temperature is 64 °C, the substrate-capillary angle 50°, the distance capillary-surface about 100 μm. The electron beam is perpendicular to the sample surface. The base pressure of the microscope is about $5 \times 10^{-7}$ mbar, which was increased to about $5.2 \times 10^{-7}$ mbar during deposition. The fabrication of complex 3D nanoarchitectures is possible thanks to the development of pattern-generation approaches[39,65]. However, in the present work the tetrapods are grown within a heuristic approach[66]. The electron beam parameters are 20 kV for the acceleration voltage and 13 pA for the beam current. The bottom (top) branches of the tetrapod structures are printed simultaneously, so that the electron beam dwells 10 ms (17 ms) on each branch before moving to the next one. The different dwell times for the top and bottom branches is necessary due to the height-dependent replenishment time of the precursor. The lateral speed of the electron beam during writing is 2.4 nm/s. The magnetic tetrapods are prepared on lithographically pre-defined $50 \times 50$ μm$^2$ Cr (5 nm)/Au (50 nm) regions on a highly doped Si wafer to reduce charging effect, which is especially relevant for XMCD-PEEM imaging. An additional XMCD contrast observed on the substrate below tetrapods in Fig. 2e–g (see also Supplementary Section 5) corresponds to the signal coming from the disk-shaped region of co-deposited $Co_3Fe$ alloy due to the electron beam scattering during the FEBID process[17]. In total, three tetrapods were studied in this work, which were fabricated with different cross-sections and opening angles between the top and bottom segments, see Supplementary Section 3. Additional semi-pyramid, diamond and cube wireframes are on a non-magnetic 1.5 μm tall PtC pillar prepared by FEBID. This elevation removes the overlapping between a shadow XMCD-PEEM contrast of the wireframe geometry and an additional contrast obtained from the co-deposited magnetic layer on the flat substrate. This enables full magnetic characterization of wireframe geometries through a set of azimuthal scans, see Fig. 5.

### XMCD-PEEM measurements

The FEBID-grown $Co_3Fe$ wireframes are investigated by means of shadow contrast imaging using the XMCD-PEEM technique[28,67,68]. The measurements are carried out at BESSY II (beamline UE49-PGM, Helmholtz-Zentrum Berlin, Germany). The experiment is done at 10 keV to minimize the risk of discharges. Reducing the extraction voltage from 20 keV to 10 keV lowers the spatial resolution by a factor of 1.4, which is partially compensated by the better energy filtering and therefore lower chromatic aberration. Thus, the resulting spatial resolution of down to 30 nm is achieved in this imaging mode.

In the experimental set-up, a circularly polarized X-ray beam irradiates the sample at a shallow angle of 16°, which introduces formation of stretched X-ray shadows being 6.7 μm long for the specific case of the considered tetrapod geometry. This shadow imaging enhances the spatial resolution along the projection direction for the magnetic state reconstruction and allows to eliminate PEEM distortions caused by the complex geometry upon direct imaging of the tetrapod. The magnetic imaging is performed at different azimuthal angles of the sample rotation with respect to the incident X-ray beam. Considering the high symmetry of the here studied wireframe geometries, it was sufficient to collect magnetic contrast at three azimuthal angles ($\phi = -45°; +45°; +135°$) to reconstruct magnetic states. We perform imaging at the $L_3$ X-ray absorption edge of Cobalt (Supplementary Section 5). The contrast of the XMCD-PEEM signal represents the projection of the magnetization distribution on the X-ray beam. Namely, parallel, perpendicular and antiparallel alignment of the magnetization with respect to the beam direction is encoded by red-white-blue color scheme, respectively. The magnetic states are imaged at remanence after fabrication of magnetic wireframes.

### Micromagnetics

Full-scale micromagnetic simulations are performed for the experimental tetrapod geometry by means of a finite-element micromagnetic (FEM) code, the successor of the GPU accelerated TETRAMAG[69,70]. The experimental geometry is reconstructed from a series of high-resolution scanning electron microscopy images obtained for different azimuthal angles, see Fig. 2b–d, based on the sculpting method in the Blender software with all spatial features of the tetrapod reflected in the final geometry. The resulting mesh is constructed from tetrahedron elements with an average size of 4.9 nm and volume of 27.2 nm$^3$. The simulations are done for a magnetic object with micromagnetic parameters of $Co_3Fe$ polycrystalline media: saturation magnetization $\mu_0 M_s = 1.88$ T, where $\mu_0$ is the vacuum permeability, exchange constant $A = 14$ pJ/m and exchange length $\ell = \sqrt{2A/(4\pi M_s^2)} = 3.2$ nm. As polycrystalline $Co_3Fe$ does not have strong crystalline magnetic anisotropy, the micromagnetic description

considers only the exchange and magnetostatic contributions to the total magnetic energy:

$$E_{TOT} = E_{EX} + E_{MS}, \tag{2}$$

$$E_{EX} = A \int_V d\boldsymbol{r} \left[ (\nabla m_x)^2 + (\nabla m_y)^2 + (\nabla m_z)^2 \right], \tag{3}$$

$$\frac{E_{ms}}{M_s^2} = \frac{1}{2} \iint_{SS} dSdS' \frac{\varsigma(\mathbf{r})\varsigma(\mathbf{r}')}{|\mathbf{r}-\mathbf{r}'|} + \frac{1}{2} \iint_{VV} d\mathbf{r}d\mathbf{r}' \frac{\lambda(\mathbf{r})\lambda(\mathbf{r}')}{|\mathbf{r}-\mathbf{r}'|} + \iint_{VS} d\mathbf{r}dS' \frac{\lambda(\mathbf{r})\varsigma(\mathbf{r}')}{|\mathbf{r}-\mathbf{r}'|}, \tag{4}$$

where $A$ is the exchange constant, $\boldsymbol{m} = \boldsymbol{M}/M_s$ is the normalized magnetization with $M_s$ being the saturation magnetization, $\varsigma(\mathbf{r}) = \mathbf{m}(\mathbf{r}) \cdot \mathbf{n}(\mathbf{r})$ and $\lambda(\mathbf{r}) = -\nabla \cdot \mathbf{m}(\mathbf{r})$ are surface and volume magnetostatic charges, respectively. The equilibrium magnetic states for the tetrapod geometries are obtained relying on the energy minimization approach by means of a conjugate gradient method starting from the initial states corresponding to a homogeneously magnetized object along the $\hat{\boldsymbol{x}}, \hat{\boldsymbol{y}}, \hat{\boldsymbol{z}}$ directions, and artificially defined domain wall and antivortex states in the conjugation point of the four wireframe branches. The calculations are carried out for experimentally reconstructed tetrapod shape and other models of wireframe structures discussed in the main text and Supplementary Section 2, 6–8. In the idealized models, we realized intersections of magnetic segments with circular cross-section and rounded ends.

For investigations of the stray fields, we conduct additional calculations of stray field distributions around the tetrapods with equilibrium magnetic states obtained micromagnetically (Supplementary Section 8). Namely, every studied geometry is surrounded by a $6 \times 6 \times 6 \, \mu m^3$ "airbox", which represents the outer non-magnetic space and has its own unstructured FEM discretization compatible with the tetrapod mesh. In the following, a stray field calculation $\boldsymbol{H}(\boldsymbol{r}) = -\nabla \psi(\boldsymbol{r})$ is done by using the magnetostatic scalar potential $\psi(\boldsymbol{r})$ obtained from the Poisson equation[71,72]

$$\nabla^2 \psi(\boldsymbol{r}) = \begin{cases} \nabla \cdot \boldsymbol{m}(\boldsymbol{r}) & \text{if } \boldsymbol{r} \in V, \\ 0 & \text{outside the tetrapod}, \end{cases} \tag{5}$$

with the corresponding boundary conditions at the volume surface $S$

$$\psi_{in}(\boldsymbol{r})|_S = \psi_{out}(\boldsymbol{r})|_S, \quad \frac{\partial \psi_{in}(\boldsymbol{r})}{\partial \boldsymbol{n}(\boldsymbol{r})}\bigg|_S - \frac{\partial \psi_{out}(\boldsymbol{r})}{\partial \boldsymbol{n}(\boldsymbol{r})}\bigg|_S = \sigma(\boldsymbol{r})|_S, \tag{6}$$

where $\sigma(\boldsymbol{r}) = \boldsymbol{m}(\boldsymbol{r}) \cdot \boldsymbol{n}(\boldsymbol{r})$ is a surface magnetostatic charge. The corresponding calculation of Eq. (5) with the finite element boundary conditions (Eq. (6)) is done by means of the Fredkin-Koehler method[73].

## Optimization of the tetrapod geometry

We performed full-scale micromagnetic simulations of seven different models of tetrapods (Supplementary Section 2). Each model is constructed from four nanowires of different diameter, length, opening and rotation angles. We note that although the geometries are different, all tetrapods shown in Supplementary Section 2 are homeomorphic (i.e., they are topologically identical). By changing the rotation angle between top and bottom parts of the tetrapod, we obtain an important transition from the $D_{4h}$ (plane tetrapod) to the $T_d$ (tetrapod with 90° rotation of the top part) symmetry point groups, which leads to the corresponding design of shape anisotropy and introduces a tuning knob for spin textures. Relying on the results obtained by the Γ-convergence methods[74–76], for the case of a flat

tetrapod we conclude that the principal axis is the hard axis of magnetization. Accordingly, by virtue of the arguments given in[77] and based on the energy-conditioned order parameter space, we conclude that the stability of the bulk antivortex is supported by the non-trivial fundamental group, $\pi_1(\mathbb{S}^2 \setminus \{-\hat{\boldsymbol{y}}, \hat{\boldsymbol{y}}\}) = \pi_1(\mathbb{S}^1 \times I) = \pi_1(\mathbb{S}^1) = \mathbb{Z}$. When the shape of the tetrapod changes from flat to volumetric, the above axis of hard magnetization disappears, and, accordingly, the vortex loses its topology-based stability.

Simulations for ideal symmetric tetrapod geometries (Fig. 1i) show that the bulk antivortex state has lower energy than the surface states in the whole range of the rotation angles $\alpha$. Still, near $\alpha = 90°$ the surface and bulk textures have similar energies. The corresponding range of geometric shapes can be substantially increased by introducing additional asymmetries, e.g., by different line segment radii (Fig. 1j). Thus, as the states are of similar energy, running magnetic hysteresis can stabilize the state of interest in magnetic tetrapods of appropriate geometry. We note that the surface state stabilized in this way will be an equilibrium state, yet not necessarily the ground state. Our simulations do not exclude the possibility that different geometry of a tetrapod may make the energy of the surface state lower than that of the bulk. In this respect, the experimental geometry has small differencies even compared to the customized asymmetric tetrapod in Fig. 1j due to fabrication imperfections. Although it does not seem possible to strictly determine whether the surface state is the ground state for the experimental system, it is the only observed state in the corresponding simulations.

## Determination of the position of magnetic solitons based on the topological charge density

Following the approach[29], we analyse the distribution of the flux density of the topological charge[78–81] over the 3D geometry:

$$\Omega_l = \frac{1}{8\pi} \epsilon_{lno} \epsilon_{ijk} m_i \partial_n m_j \partial_o m_k, \tag{7}$$

where $m_i$ is the normalized local magnetization component, $\epsilon_{ijk}$ is the Levi–Civita tensor and $i, j, k, l, n, o = \{x, y, z\}$. Being the topological density of the second homotopy group, $\Omega$ is suitable for recognizing vortices as well. The latter is due to the fact that magnetic vortices with a magnetization lying in $\mathbb{S}^2$ are merons[82]. Thus, the maximum of $\Omega$ unambiguously determines the position of surface topological magnetic solitons in the vicinity of the surface and Bloch lines in the interior of arbitrarily bent samples. As $\Omega$ is dependent on the type of the magnetic texture, it is more convenient to introduce the normalized flux density of the topological charge $\widetilde{\Omega} = \Omega/\Omega_{max}$, where $\Omega_{max}$ is the absolute maximum value of $\Omega$ for the particular magnetization distribution. The applied analysis approach could be extended by the study of emergent magnetic fields[83], $\boldsymbol{B}^e = 4\pi\hbar \, \boldsymbol{\Omega}$. This method is suitable for the calculation of the Berry phase and corresponding topological Hall effect[84].

## Surface vs. bulk magnetic textures

Topological properties and the description of vector fields are dependent on the dimensionality of the space under consideration. In this work, we rely on the Poincaré–Hopf theorem. Therefore, our discussion is limited to isolated (point-like) topological defects (i.e., vortices or antivortices) in 2D compact manifolds immersed into 3D space. The role of compactness, i.e., the absence of a boundary of a manifold, can be illustrated as follows. In an ultrathin circular nanodot, a vortex is a purely 2D magnetic texture. In this case, the difference between the nanodot and extended film is given by the presence of an edge around the nanodot. The edge results in the stabilization of the vortex texture due to magnetostatic interaction. Other magnetic states (e.g., onion or C-states) emerge as a result of the competition between magnetostatics and exchange. These states can be understood as boundary states.

For a nanodot of a finite thickness (cylinder), the geometric topological counterpart is a ball instead of an extended film. According to the hairy ball theorem, any vector field on the surface of a ball (this surface has no edges) must have topological defects possessing the total vorticity $Q^\Sigma = +2$. This total vorticity can be realized by the formation of (i) vortices (in this case, two isolated topological defects are located at the top and bottom cylinder surfaces) or (ii) onion or C-states (in this case, topological defects are located on the side face of the cylinder).

The Poincaré–Hopf theorem can be applied also for the case of non-isolated topological defects[85]. This discussion is irrespective of the bulk continuation of the magnetic textures observed at the surface because the behavior of the interior magnetic texture is driven by the whole geometry, such as transformation of the bulk antivortex with a Bloch line to a pair of two surface antivortices.

The presented discussion can be extended to the properties of bulk topological textures characterized by a nonzero Hopf index[86,87]. However, any 3D manifold hosting a magnetization vector field has natural boundaries and the corresponding topological density for one-point compactification of $\mathbb{R}^3$ can give non-integer values being integrated over the sample volume[88].

## Data availability
The data supporting the findings of this study are available within the main text of this article and its Supplementary Information. Further information on this study is available from the corresponding authors upon reasonable request. Source data are provided with this paper.

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

## Acknowledgements

We thank L. Bischoff (HZDR) for his support with SEM measurements of tetrapod structures and I. Antonenko (HZDR) for the support in the reconstruction of the shape of the experimentally studied tetrapod. Support by the Ion Beam Center large-scale facilities at the HZDR is gratefully acknowledged. XMCD-PEEM measurements were carried out at the SPEEM instrument (UE49 PGMa) at the BESSY II electron storage ring operated by the Helmholtz-Zentrum Berlin für Materialien und Energie. This work was financed in part via European Union's Horizon Europe Research and Innovation Programme, Grant Agreement No. 101070066 (project REGO) and the German Research Foundation (DFG) Grants No. MA5144/22-1, MA5144/24-1, HU752/16-1, BA6595/1-1, VO2598/1-1. S.L.C. acknowledges financial support by the Vienna Doctoral School in Physics (VDSP). S.L.C. and O.V.D. acknowledge support by the Austrian Science Fund (FWF) under Grant No. I 4889 (CurviMag). S.L.C. acknowledges European Cooperation on Science and Technology (e-COST) Action CA19108 (Hi-SCALE) for support via the STSM E-COST-GRANT-CA19108-2d219934. F.N.R. acknowledge support from the Swedish Research Council. O.E. also acknowledge support from the European Research Council via Synergy Grant No. 854843 (the FASTCORR project), the Wallenberg Initiative Materials Science for Sustainability (WISE) funded by the Knut and Alice Wallenberg Foundation (KAW), eSSENCE, and STandUP.

## Author contributions

FP, AlK, SB, MH designed, optimized fabrication conditions and fabricated tetrapod structures with the support of SLC and OVD. XMCD-PEEM studies were done by FK, MAM, OMV, PM, SS with the support of DM and AlK. FNR, OE, OMV, OVP, DM performed the symmetry analysis of magnetic geometries. AK developed the micromagnetic code to perform calculations showed in this work. Micromagnetic modeling was carried out by OMV, AK. The concept of high-order vorticity in magnetic wireframes was developed by OVP, DM, OMV, JAFR with support of FP, FK, AK, MH, OVD. The study and analysis of magnetic stray fields was carried out by JAFR, OVP, OMV, DM, AK with the support by SLC, OVD, MH. The manuscript was written by OMV, OVP, DM, JAFR with contribution from MH, FP, FK, AK, FNR, OE, OVD. All co-authors contributed to the revisions of the manuscript and discussions of the manuscript results. The project was supervised by DM, MH, OVD, FK.

## Funding

## Competing interests

The authors declare no competing interests.
