## [Peer Review File · Nature Communications]

Reviewers' Comments:

Reviewer #1:

Remarks to the Author:

The authors report on three-dimensional magnetic textures with high order vorticity in soft magnetic wireframes. Importantly, the authors consider a multi-pod system with m holes, and the relationship between the Euler characteristic χ in the manifold and the total vorticity Q^Σ of this vector field is determined by Poincaré–Hopf theorem. As a result, the shape supports N vortices and $N-\chi$ antivortices in equilibrium, providing $2N-\chi$ topological solitons in total. The predictions are borne out in micromagnetic simulations and experiments.

This is a timely paper because topologically nontrivial magnetic state is at the very heart of modern magnetism/topology community. I believe that this paper has the potential to be met with interest not only by the magnetism community but also by the general condensed matter community. On other hand, a few things and problems need to be clarified by the authors before I can recommend the acceptance of the manuscript for Nature Communications:

1. The details of simulations and the corresponding model Hamiltonian should be provided.
2. I agree with the authors that a system containing many vortexes can effectively mediate the transport of spin waves, and provides a promising platform to achieve 3D spin-wave splitter. Specifically, when a single spin wave packet propagates through a vortex, it will experience an effective Lorentz force, resulting in the skew scattering of magnons [Phys. Rev. B 107, 214418 (2023)]. On the other hand, in a vortex network including many domain walls, the spin waves can propagate along these nanochannels [Phys. Rev. Lett. 114, 247206 (2015)]. However, in this work, the vortexes placed at the connection area and the ends of the tetrapod segments with far distance, which limits its application on magnonic devices. Can the distance between them be shortened in this 3D soft magnetic wireframe structures?
3. Can the authors briefly envision the effect of opening angle between two top/bottom segments in the tetrapod geometry on the magnetic texture?
4. The dipolar interaction plays an important role in stabilizing magnetic textures, which can be mediated by the size of system. Correspondingly, when the diameter of tetrapod segments increases, what is the effect of the changed magnetic interaction on the number and size of magnetic vortexes?
5. It is well known that spatial curvature always induces an effective Dzyaloshinskii-Moriya interaction (DMI) in the curved surfaces [Phys. Rev. Lett. 120, 067201 (2018)]. In this system, can skyrmions emerge on the surfaces with finite spatial curvature?

Reviewer #2:

Remarks to the Author:

The authors present a very interesting work on the introduction of magnetic topological textures via pure geometric design of 3D wire-frame magnetic architectures. Authors exploit geometrical topology to control the magnetisation field topology, demonstrating it with an elegant experiment. This introduces an excellent platform to perform fundamental studies related with magnetic topological solitons in a very deterministic and controlled manner. Introduction, hypothesis, experiment and modelling seems to be performed adequately, and the conclusions are supported by the obtained results.

I would recommend its publication in Nature Communications after a minor revision of some points and some clarifications. Here follow my comments:

1) The computational study performed in the magnetic tetrapod analysing the twist angle between bottom and top parts mentions that the jump in total winding number appears above 89 degrees of twist. Is this really 89 or it trends asymptotically to 90?

If the stable energetic configuration for a resulting winding number is 90 degrees only, isn't it surprising that the fabricated structure presents that state? Has any magnetic protocol been realised to reach this state? Authors should comment about it.

2) In page 3 when discussing the possibilities to use this structures for reservoir/probabilistic computing, I would appreciate a clarification about how the complexity necessary for these approaches could be achieved in a system so deterministic due to its well defined geometry and

texture localization and stability. In fact, although the number of textures is as the author's state virtually infinite, you need to add as many rods and crosslinks that you need in order to achieve a large enough base of defects to perform reservoir computing. Moreover, how is the size of the knot affecting the formed textures as well as their location/magnetic connection is not clear.

3) In page 5, the authors state that they are using for their micromagnetic simulations the experimentally reconstructed geometry of the tetrapod but no indication on how they have achieved this is mentioned neither in the methods, nor in the main text of the manuscript. It is only mentioned that high resolution SEM images were used. It would be good to indicate if this has been performed using any kind of 3D structural reconstruction method such as photogrammetry and add the adequate references for the method/code used. I would appreciate also a comment about how the authors have addressed the issue of image alignment, if it is photogrammetry, when dealing with FEBID structures with almost no features in the surface of the structure.

4) Also in page 5, authors make the discussion about the connection between the two surface anti-vortices based on the absence of a Bloch line connecting their cores. This is also the approach followed for the rest of analysed textures. Wouldn't be interesting to show this connection/isolation by representing/analysing the magnetic vorticity/gyrovector/magnetic emergent field of the magnetization? This is a very powerful method that allows to observe how different topologically interesting items within a magnetic system are linked as previously used in "C. Donnelly et al Nature Physics 17, pages 316–321 (2021)" or in "J. Hermosa et al Communications Physics 6, 49 (2023)". This representation could be excellent for the applications indicated by the authors related with magnonics and spin-wave propagation through domain walls and Bloch lines.

5) Imaging conditions of the PEEM are not fully indicated in the methods. I mean the acceleration voltage for the extraction of photoelectrons. Was it 10 keV or 20 keV or other? How this affected the measurements in terms of discharges/resolution conditions?

6) In figure 4, although the structures are claimed to be fabricated on top of a non-magnetic pillar, there is XMCD signal in the base of the structures showing also a vortex-compatible signal. Could the authors comment about those features and their potential impact on the final magnetic state of the system?

7) Again about these more complex and N-torus like structures, how are the magnetic states imaged obtained? Are the samples in the as-grown state? Could the magnetic state present in the samples be controlled by the fabrication order of the rods of the wireframe due to magnetostatic interactions between the rods?

8) In figure 5, the panel a) is under my point of view somehow confusing. Here, the streamlines following the stray field of the structure have a rainbow colormap with associated to the B_z component intensity but the colormap scale is missing. Although one can reconstruct its meaning from the magnetic configuration in the tetrapod, I think it would be easier for the reader to have the colormap scale pointing out what means positive and negative, as well as more and less intense.

9) The discussion about the so called "Complex stray field texture emerging from the anti-vortices" is quite obscure as it is not clearly observed from the central cross section representation. This is I think because the authors want to explain the rotation of the field due to the geometry of the tetrapod at the same time that they point out this field texture around the antivortices at the connection. I would separate both discussions and magnify the area around the antivortices to favour the identification, discussion and understanding of this mentioned texture.

10) To finish with figure 5, the solid colours selected for the representation of the B_z/B_{zmax} isosurfaces are confusing in terms of the rainbow colormap which appears in figure 5 panel k). The colours of the isosurfaces does not correspond to the levels shown in the colormap scale.

Response letter

We thank the Reviewers for providing their constructive remarks that helped us improving the clarity of the manuscript. Our itemized responses to the remarks of the Reviewers are listed below. The manuscript has been substantially revised and restructured, especially in the Supplementary Information. Therefore, only major changes in the manuscript are indicated in blue.

Reviewer #1

Comment 1:

The authors report on three-dimensional magnetic textures with high order vorticity in soft magnetic wireframes. Importantly, the authors consider a multi-pod system with m holes, and the relationship between the Euler characteristic χ in the manifold and the total vorticity Q^Σ of this vector field is determined by Poincaré–Hopf theorem. As a result, the shape supports N vortices and $N - \chi$ antivortices in equilibrium, providing $2N - \chi$ topological solitons in total. The predictions are borne out in micromagnetic simulations and experiments.

This is a timely paper because topologically nontrivial magnetic state is at the very heart of modern magnetism/topology community. I believe that this paper has the potential to be met with interest not only by the magnetism community but also by the general condensed matter community. On other hand, a few things and problems need to be clarified by the authors before I can recommend the acceptance of the manuscript for Nature Communications.

Answer:

We thank the Reviewer for his/her positive assessment of our work and recommendation of its publication in Nature Communications. The manuscript has been revised according to the suggestions of the Reviewer.

Comment 2:

The details of simulations and the corresponding model Hamiltonian should be provided.

Answer:

Relevant details of the micromagnetic model and simulations have been added to the Methods section of the manuscript.

The following text has been added in the **Micromagnetics** section in Methods (page 15):

As polycrystalline Co_3Fe does not have strong crystalline magnetic anisotropy, the micromagnetic description considers only the exchange and magnetostatic contributions to the total magnetic energy:

$$E_{\text{TOT}} = E_{\text{EX}} + E_{\text{MS}}, \quad (\text{R1})$$

$$E_{\text{EX}} = A \int_V d\mathbf{r} [(\nabla m_x)^2 + (\nabla m_y)^2 + (\nabla m_z)^2], \quad (\text{R2})$$

$$\begin{aligned} \frac{E_{\text{ms}}}{M_s^2} = & \frac{1}{2} \iint_{SS} dS dS' \frac{\zeta(\mathbf{r}) \zeta(\mathbf{r}')}{|\mathbf{r} - \mathbf{r}'|} + \frac{1}{2} \iint_{VV} d\mathbf{r} d\mathbf{r}' \frac{\lambda(\mathbf{r}) \lambda(\mathbf{r}')}{|\mathbf{r} - \mathbf{r}'|} \\ & + \iint_{VS} d\mathbf{r} dS' \frac{\lambda(\mathbf{r}) \zeta(\mathbf{r}')}{|\mathbf{r} - \mathbf{r}'|}, \end{aligned} \quad (\text{R3})$$

where A is the exchange constant, $\mathbf{m} = \mathbf{M}/M_s$ is the normalized magnetization with M_s being the saturation magnetization, $\zeta(\mathbf{r}) = \mathbf{m}(\mathbf{r}) \cdot \mathbf{n}(\mathbf{r})$ and $\lambda(\mathbf{r}) = -\nabla \cdot \mathbf{m}(\mathbf{r})$ are surface and volume magnetostatic charges, respectively.

Comment 3:

I agree with the authors that a system containing many vortexes can effectively mediate the transport of spin waves, and provides a promising platform to achieve 3D spin-wave splitter. Specifically, when a single spin wave packet propagates through a vortex, it will experience an effective Lorentz force, resulting in the skew scattering of magnons [Phys. Rev. B 107, 214418 (2023)]. On the other hand, in a vortex network including many domain walls, the spin waves can propagate along these nanochannels [Phys. Rev. Lett. 114, 247206 (2015)]. However, in this work, the vortexes placed at the connection area and the ends of the tetrapod segments with far distance, which limits its application on magnonic devices. Can the distance between them be shortened in this 3D soft magnetic wireframe structures?

Answer:

We thank the Reviewer for this insightful remark. Indeed, in contrast to planar rectilinear systems where spin-waves could be channelled along narrow domain walls or homogeneously magnetized stripe channels, we envision the possibility to utilize linear segments of complex wireframe geometries as spin-wave channels. For instance, high-frequency modes could propagate along homogeneously magnetized segments and interact with vortex and antivortex textures.

Following the remark of the Reviewer, we perform additional full-scale micromagnetic simulations to address the question of how close magnetic topological textures could be located at tetrapod geometries. In this respect, we simulated magnetic states of tetrapods with different length of nanowires. In particular, the length of all linear segments forming the tetrapod geometry was shortened from the initial length of $1.3 \mu\text{m}$ to 300 nm with a step of 200 nm . We track changes in the magnetic stream lines, see Figure R1(a–c). For the case of the longest segments, the magnetic state within segments becomes uniform when approaching the central part of the sample. This is not the case for narrower segments, where the vortex core approaches the central region with antivortices. For the smallest sample with 300 nm -long segments, the interaction between textures is strong enough to induce an additional vortex-antivortex pair still maintaining the total vorticity $+2$.

The following text is added to the Supplementary Information (page 4), section on the **Impact of the tetrapod geometry on magnetic textures**:

Figure R1. Effect of the line segment length of the tetrapod geometry on the magnetic texture. (a – c) Equilibrium magnetic states of tetrapods constructed from 300 nm, 700 nm and 1.1 μm long line segments. Green regions inside each geometry represent the isosurface of the topological charge density distribution, that reach their maxima near the topologically non-trivial textures. Arrows show magnetization stream-lines of the magnetic states. (d – f) Angular projections of the surface magnetization distribution shown in panels (a – c), respectively, on an elemental sphere. Here, (θ, ϕ) are azimuthal and polar coordinates. Green circles indicate the position of vortices and orange diamonds show the position of antivortices. The radius of the line segments forming the tetrapod geometries studied here is 58 nm, the rotation angle $\alpha = 90^\circ$ and the opening angle $\beta = 90^\circ$.

Length of the tetrapod line segments

To investigate the influence of shortening of the tetrapod line segments on the equilibrium magnetic states, we perform micromagnetic simulations for geometries constructed from rounded linear nanowires of different length, $L \in [300; 500; 700; 900; 1100; 1300]$ nm, with the same radius $r = 57$ nm, the rotation angle $\alpha = 90^\circ$ and opening angle $\beta = 90^\circ$. In the case of the tetrapod with the shortest line segments ($L = 300$ nm), the resulting equilibrium magnetic state possesses an additional vortex-antivortex pair in the central region due to the formation of the bulk vortex Bloch lines that intertwine in the connection area, see Supplementary Fig. R1(a). Thus, the surface magnetization distribution acquires 5 vortices and 3 antivortices, resulting in the total vorticity $Q^\Sigma = +2$ being in line with the Euler characteristic of the geometry $\chi = +2$, see Supplementary Fig. R1(c). With the elongation of the line segments to $L = 700$ nm, the vortex structure appears only as a surface state at their ends without the formation of Bloch lines along segments, see Supplementary Fig. R1(b). This results in the emergence of a pair of antivortices at the surface of the connection region despite the magnetization curling in linear segments near the connection area, see Supplementary Fig. R1(e). Further elongation of the line segments to $L = 1.1 \mu\text{m}$ does not result in a change of the total number of magnetic textures in the tetrapod, but it leads to the formation of homogeneously magnetized linear segments in the central area of the tetrapod with a reduction of the curling of magnetization at the surface, see Supplementary Fig. R1(c) and (f). We note that the segment length determines the strength of the interaction between vortices at the segment ends and antivortices in the central part of the geometry. If the segments are short enough, the vortex cores can substantially affect antivortices in the center, even leading to the creation of an additional vortex-antivortex pair in equilibrium.

Comment 4:

Can the authors briefly envision the effect of opening angle between two top/bottom segments in the tetrapod geometry on the magnetic texture?

Answer:

To address this remark of the Reviewer, we performed additional micromagnetic simulations for tetrapods with the rotation angle $\alpha = 90^\circ$ between the planes accommodating the top and bottom pairs of nanowires forming the tetrapod geometry. In these additional simulations, we varied the opening angle β between two top/bottom segments in the tetrapod. The simulations reveal that the change of the opening angle affects the equilibrium position of two antivortices, while the general topology of the vector field on the surface of the tetrapod is preserved with the total vorticity being equal to $+2$. Namely, for tetrapods with the opening angle of 30° the pair of antivortices is positioned along the \hat{z} -axis due to the prevailing shape anisotropy. In the case of a wider opening angle, both textures are positioned in the $\hat{x}\hat{y}$ plane.

The following text is added to the Supplementary Information (page 2), section on the **Impact of the tetrapod geometry on magnetic textures**:

Figure R2. Effect of the opening angle on magnetic states in tetrapods. (a) The tetrapod geometry with an opening angle $\beta = 30^\circ$ leads to the effective elongation of the central area along the \hat{z} -axis. Green regions correspond to the isosurfaces in the distribution of the topological charge density $\tilde{\Omega} = 0.1$. Maxima in these distributions correspond to the locations of topologically non-trivial textures. Stream-arrows depict the magnetization state inside the tetrapod. (b) Mapping of the surface magnetization distribution of the tetrapod onto a unit sphere. (c) Angular projection of the sphere in azimuthal and polar coordinates, (θ, ϕ) . Panels (d – f) and (g – i) show the corresponding information for the tetrapods with the opening angle $\beta = 60^\circ$ and $\beta = 90^\circ$, respectively. The length of the line segments forming the here studied tetrapod geometries is $L = 1.3 \mu\text{m}$, the radius of line segments is $r = 57 \text{ nm}$, rotation angle $\alpha = 90^\circ$.

Opening angle

The opening angle β is defined as the angle between the bottom segments of the tetrapod; the angle between the top segments of the tetrapod is kept the same. We perform micromagnetic simulations for the tetrapod geometries with the rotation angle $\alpha = 90^\circ$ and constructed from the $1.3 \mu\text{m}$ long linear nanowires with the radius of 57 nm and opening angles $\beta \in [30^\circ; 45^\circ; 60^\circ; 75^\circ; 90^\circ]$. Starting from different initial configurations, we stabilize various equilibrium magnetic textures presented in Supplementary Fig. R2. In the case of opening angle $\beta = 30^\circ$, the equilibrium state acquires the total vorticity $Q^\Sigma = +2$, with four vortices and two antivortices. An additional shape anisotropy that appears in the central tetrapod region, due to the spatial elongation, leads to the formation of a hybrid vortex-antivortex state with two antivortices at the surface being positioned along the \hat{z} -axis, and a bulk vortex Bloch line inside the central region. For geometries with the opening angle β larger than 45° , both antivortex states are positioned in the $\hat{x}\hat{y}$ plane of the central tetrapod region.

Comment 5:

The dipolar interaction plays an important role in stabilizing magnetic textures, which can be mediated by the size of system. Correspondingly, when the diameter of tetrapod segments increases, what is the effect of the changed magnetic interaction on the number and size of magnetic vortices?

Answer:

Motivated by this comment of the Reviewer, we performed additional simulations to investigate how the size and diameter of line segments forming the tetrapod geometry affect the resulting equilibrium magnetic states. In particular, we simulate tetrapod geometries constructed from (i) nanowires of the same radius $r = 57 \text{ nm}$ but different lengths $L = [300; 500; 700; 900; 1100; 1300] \text{ nm}$ and (ii) the same length $L = 1.3 \mu\text{m}$ but different radii $r = [10; 20; 30; 40; 50; 60; 70; 80; 90; 100] \text{ nm}$. The first series of simulations

Figure R3. Effect of the radius of line segments forming the tetrapod geometry on magnetic textures. (a – e) Equilibrium magnetization states of the tetrapods with $r = [10; 30; 50; 70; 90]$ nm, respectively. Green regions correspond to the isosurfaces in the distribution of the topological charge density $\tilde{\Omega} = 0.1$. Maxima in these distributions correspond to the locations of topologically non-trivial textures. Stream-arrows depict the magnetization state inside the tetrapod. **(f – j)** Angular projections of the surface magnetization distributions shown in panels (a – e), respectively. Green circles indicate the position of vortices and orange diamonds show the position of antivortices. The length of the line segments forming the here studied tetrapod geometries is $L = 1.3 \mu\text{m}$, the rotation angle $\alpha = 90^\circ$ and opening angle $\beta = 90^\circ$.

reveals the formation of an additional vortex-antivortex pair in the central region of the tetrapod. This texture appears as a result of the vortex Bloch lines entering from the nanowire ends to the central area of the tetrapod. The same effect happens in the case of the tetrapod geometry constructed from wide linear segments: the vortex Bloch lines become longer and interweave in the central region.

The following text is added to the Supplementary Information (page 3), section on the **Impact of the tetrapod geometry on magnetic textures**:

Length of the tetrapod line segments

To investigate the influence of shortening of the tetrapod line segments on the equilibrium magnetic states, we perform micromagnetic simulations for geometries constructed from rounded linear nanowires of different length, $L \in [300; 500; 700; 900; 1100; 1300]$ nm, with the same radius $r = 57$ nm, the rotation angle $\alpha = 90^\circ$ and opening angle $\beta = 90^\circ$. In the case of the tetrapod with the shortest line segments ($L = 300$ nm), the resulting equilibrium magnetic state possesses an additional vortex-antivortex pair in the central region due to the formation of the bulk vortex Bloch lines that intertwine in the connection area, see Supplementary Fig. R1(a). Thus, the surface magnetization distribution acquires 5 vortices and 3 antivortices, resulting in the total vorticity $Q^\Sigma = +2$ being in line with the Euler characteristic of the geometry $\chi = +2$, see Supplementary Fig. R1(c). With the elongation of the line segments to $L = 700$ nm, the vortex structure appears only as a surface state at their ends without the formation of Bloch lines along segments, see Supplementary Fig. R1(b). This results in the emergence of a pair of antivortices at the surface of the connection region despite the magnetization curling in linear segments near the connection area, see Supplementary Fig. R1(e). Further elongation of the line segments to $L = 1.1 \mu\text{m}$ does not result in a change of the total number of magnetic textures in the tetrapod, but it leads to the formation of homogeneously magnetized linear segments in the central area of the tetrapod with a reduction of the curling of magnetization at the surface, see Supplementary Fig. R1(c) and (f). We note that the segment length determines the strength of the interaction between vortices at the segment ends and antivortices in the central part of the geometry. If the segments are short enough, the vortex cores can substantially affect antivortices in the center, even leading to the creation of an additional vortex-antivortex pair in equilibrium.

Radius of line segments

To study the influence of the radius of line segments forming the tetrapod geometry on the equilibrium magnetic states, we construct tetrapods with $L = 1.3 \mu\text{m}$ long line segments, the opening angle $\beta = 90^\circ$ and the rotation angle $\alpha = 90^\circ$, but with different nanowire radii $r \in [10; 20; 30; 40; 50; 60; 70; 80; 90; 100]$ nm. For all constructed geometries, we observe the formation of antivortex states in the central region of the tetrapod at equilibrium, see Supplementary Fig. R3. In the case of the tetrapod constructed from narrow line segments with $r \leq 30$ nm, the line segments are homogeneously magnetized. The topological defect in the magnetic texture is presented by the out-of-surface magnetization in the flower state rather than the classical vortex distribution. Still, this state topologically equivalent to the vortex one as it is shown by its angular projection of the surface magnetization on the elemental sphere, see Fig. R3f. The increase of the radius of the line segments ($40 \leq r \leq 70$ nm) leads to the formation of a surface vortex distribution at their ends. For the case of line segments with $r \geq 80$ nm, the surface vortices fill the entire volume of nanowires with the vortex Bloch line. The formation of the bulk vortex Bloch lines may potentially introduce additional vortex-antivortex surface pairs in the central region of the tetrapod.

Comment 6:

It is well known that spatial curvature always induces an effective Dzyaloshinskii-Moriya interaction (DMI) in the curved surfaces [Phys. Rev. Lett. 120, 067201 (2018)]. In this system, can skyrmions emerge on the surfaces with finite spatial curvature?

Answer:

We thank the Reviewer for this insightful remark. The formation of curvature-induced skyrmions in

geometrically curved magnetic nanoshells requires the presence of easy-normal anisotropy, see Ref.,^{1,2} which is responsible for the stabilization of out-of-surface states both in the centre of a skyrmion core and far from it. In the here discussed material system of soft polycrystalline Co_3Fe alloy in the form of magnetic nanowires (but not nanoshells), there is only shape-induced magnetic anisotropy, which does not allow to stabilize a skyrmion texture. Still, we note that magnetic Co_3Fe nanoshells can be prepared on FEBID grown PtC wires³ resulting in structural asymmetry that can potentially support out-of-surface anisotropy and intrinsic surface-type DMI. These FEBID grown architectures might provide the possibility to stabilize skyrmion textures.

The following text is added to the section **Topologically non-trivial surface states in tetrapods** in the main text (page 6):

We note that the recently reported site-selective vapour deposition using FEBID³ allows the realization of magnetic nanoshells of Co_3Fe decorating curvilinear wires made of PtC. Such surface modification may allow one to obtain a easy-normal magnetic anisotropy and intrinsic Dzyaloshinskii-Moriya interaction (DMI), which could enable the investigation of curvature-induced skyrmions^{1,4} in curvilinear FEBID-fabricated nanoshells.

Reviewer #2

Comment 1:

The authors present a very interesting work on the introduction of magnetic topological textures via pure geometric design of 3D wire-frame magnetic architectures. Authors exploit geometrical topology to control the magnetisation field topology, demonstrating it with an elegant experiment. This introduces an excellent platform to perform fundamental studies related with magnetic topological solitons in a very deterministic and controlled manner. Introduction, hypothesis, experiment and modelling seems to be performed adequately, and the conclusions are supported by the obtained results.

I would recommend its publication in Nature Communications after a minor revision of some points and some clarifications.

Answer:

We thank the Reviewer for his/her positive evaluation of our work and recommendation for its publication in Nature Communications. Following the suggestion of the Reviewer, in the revised version of the manuscript we put more emphasis on the fundamental novelty of our work.

Comment 2:

The computational study performed in the magnetic tetrapod analysing the twist angle between bottom and top parts mentions that the jump in total winding number appears above 89 degrees of twist. Is this really 89 or it trends asymptotically to 90?

Answer:

We thank the Reviewer for pointing out this aspect. We rewrote the respective discussion and updated Fig. R4 of the main text to avoid confusion related to the possibility to relax different metastable states in tetrapod structures. The revised version of the manuscript is substantially extended (especially its supplementary information) offering a systematic analysis of the evolution of the bulk and surface antivortices with respect to the rotation angle of the tetrapod. Furthermore, we comment on the role of asymmetry, which is present in the experimental geometry.

We performed additional micromagnetic simulations to study the evolution of antivortex states upon rotation in an ideal tetrapod constructed from the line segments of the same radii $r = 58$ nm and an asymmetric tetrapod with different radii of the top ($r = 40$ nm) and bottom ($r = 58$ nm) line segments. This information is used to update Fig. R4. The transition between the equilibrium bulk antivortex and two surface antivortices occurs continuously by the expansion of the homogeneously magnetized Bloch line from several exchange lengths⁵ for the D_{4h} symmetry into the full volume of the connection area in the case of the T_d symmetry, see Fig. R4i. Still, tetrapods of any rotation angle α support surface antivortices ($\eta = 0$) as metastable states. The red and blue curves in Fig. R4i show the energies of the bulk and surface antivortices as functions of the rotation angle α . The bulk antivortex always has the lower energy, except the degenerate state at $\alpha = 90^\circ$, where it has the same energy as the surface one. The energies of these states linearly diverge with deviation from $\alpha = 90^\circ$. However, the latter is not the case if the top and bottom tetrapod line segments are of different radius, see Fig. R4j. For this asymmetric geometry, the

angular range, where energies of the surface and bulk antivortices are almost the same, is substantially increased, $80^\circ \lesssim \alpha \lesssim 100^\circ$.

The following modifications were done to the main text (page 2):

The transition between the equilibrium bulk antivortex state ($\eta = -1$) and two surface antivortices ($\eta = 0$) occurs continuously with the change of α , accompanied by the expansion of the homogeneously magnetized Bloch line. In particular, the characteristic size of the Bloch line changes from several exchange lengths⁵ (for the case of the D_{4h} symmetry) to consume the entire connection region (for the case of the T_d symmetry), see Fig. R4i and Supplementary Section 1. The symmetrical branches around $\alpha = 90^\circ$ are a consequence of the identical shapes and at the same time reflect the duality regarding the location of the antivortices. Only for $\alpha = 90^\circ$, states with the surface and bulk antivortices become symmetry-connected and, accordingly, equal in energy, while for all other α the bulk antivortex has slightly lower energy. An increase of the angular range where both these states have almost the same energies ($80^\circ \lesssim \alpha \lesssim 100^\circ$) can be achieved if the top and bottom tetrapod parts are asymmetric, e.g. the diameter of top and bottom line segments is different, see Fig. R4j. This type of asymmetric geometry is of advantage for experimental explorations aiming to realize surface antivortex textures. This enables lower magnetization gradients in the connection regions of the tetrapod line segments and favors the internal part of the connection region to be homogeneously magnetized.

And (page 15):

Simulations for ideal symmetric tetrapod geometries (Fig. R4i) show that the bulk antivortex state has lower energy than the surface states in the whole range of the rotation angles α . Still, near $\alpha = 90^\circ$ the surface and bulk textures have similar energies. The corresponding range of geometric shapes can be substantially increased by introducing additional asymmetries, e.g., by different line segment radii (Fig. R4j). Thus, as the states are of similar energy, running magnetic hysteresis can stabilize the state of interest in magnetic tetrapods of appropriate geometry. We note that the surface state stabilized in this way will be an equilibrium state, yet not necessarily the ground state. Our simulations do not exclude the possibility that different geometry of a tetrapod may make the energy of the surface state lower than that of the bulk. In this respect, the experimental geometry has small differences even compared to the customized asymmetric tetrapod in Fig. R4j due to fabrication imperfections. Although it does not seem possible to strictly determine whether the surface state is the ground state for the experimental system, it is the only observed state in the corresponding simulations.

In addition, we modified the Figure R4j in the main text and Figure R5a in the Supplementary Information and introduced a new section **Transformation between bulk and surface antivortices** in the Supplementary Information:

Figure R4. Magnetization mapping for compact manifolds: Soft magnetic tetrapod. (a) Schematics of an infinite easy-plane magnetic thin film with homogeneous magnetization, which can support localized vortex (V) - antivortex (AV) pairs. The vorticity of the vortex (antivortex) is $+1$ (-1). (b) Corresponding mapping of the texture shown in (a) on the unit sphere with a hole. The Euler characteristic of this geometry is $\chi = 0$. Black arrows depict the magnetization distribution on a surface. (c) Two vortices of same circulation are formed in equilibrium at the poles of a complete sphere in line with the hairy ball theorem. The Euler characteristic of a sphere is $\chi_{\text{sphere}} = +2$. (d) The stability of a complex magnetic texture with N vortices and $(N-2)$ antivortices in an N -pod and (e) its mapping onto a unit sphere. The total vorticity is $Q^\Sigma = +2 \equiv \chi_{\text{sphere}}$ as the N -pod is homeomorphic to a sphere. (f) Tetrapod geometry constructed of four line segments with a length $L = 1.6 \mu\text{m}$ and circular cross-section with radius $r = 58 \text{ nm}$. The opening angle between the two top and two bottom line segment pairs is $\beta = 90^\circ$. Each pair of line segments forms a plane. These planes are rotated by the rotation angle α with respect to each other. (g), (h) Schematic illustrations of the *bulk* and *surface* antivortex states, respectively, that form in the centre interconnection region of tetrapods with $\alpha = 0^\circ$ and $\alpha = 90^\circ$, respectively. The planes S represent the target space for the calculation of the winding number of the corresponding magnetization distributions depicted by the black arrows. (i) and (j) Dependencies of the total energy of the two states, which define *bulk* (red squares) and *surface* (blue disk) antivortices on the rotation angle α for the (i) symmetric (all line segments have the radius $r = 58 \text{ nm}$) and (j) asymmetric tetrapods (bottom line segments have $r = 58 \text{ nm}$, while top ones have $r = 40 \text{ nm}$). At $\alpha = 90^\circ$ both states coincide up to a symmetry transformation. For the asymmetric tetrapod, there is a wider area by α , where the energies of these states are almost equal.

Figure R5. Schematic image of the tetrapod geometry with two parts rotated relative to each other.

Transformation between bulk and surface antivortices

We consider the transition between the bulk and surface states in a tetrapod by varying the rotation angle α between the top and bottom parts of the tetrapod. We propose a continuous field configuration $\mathbf{m}(x', y', z')$ that represents the bulk vortex state for $\alpha = 0$ (planar tetrapod in the $x'y'$ plane) and the surface vortex state for $\alpha = \pi$ (planar tetrapod, but in the $x'z'$ plane). Let the radius of each tetrapod line segment be equal to r . The model transformation for the antivortex solution in the central region of the tetrapod can be derived by the action of the rotation matrix \mathbf{M} on a trivial collinear state, namely:

$$\mathbf{m} = \mathbf{M} \cdot \begin{pmatrix} 0 \\ 0 \\ -1 \end{pmatrix}, \quad (\text{R4})$$

with the expression for the objective matrix \mathbf{M} ,

$$\mathbf{M} = \mathbf{R}(\vartheta_1, \mathbf{k}_1) \cdot \mathbf{R}(\vartheta_2, \mathbf{k}_2), \quad (\text{R5})$$

where $\mathbf{R}(\vartheta, \mathbf{k})$ is the rotation transformation matrix that denotes rotation by the angle ϑ around a given vector $\mathbf{k} = \{k_x, k_y, k_z\}$ (the Rodrigues' rotation matrix)

$$\mathbf{R}(\vartheta, \mathbf{k}) = \mathbf{I} + \sin(\vartheta)\mathbf{K} + (1 - \cos(\vartheta))\mathbf{K} \cdot \mathbf{K}, \quad \mathbf{I} \equiv \begin{pmatrix} 1 & 0 & 0 \\ 0 & 1 & 0 \\ 0 & 0 & 1 \end{pmatrix}, \quad \mathbf{K} \equiv \begin{pmatrix} 0 & -k_z & k_y \\ k_z & 0 & -k_x \\ -k_y & k_x & 0 \end{pmatrix}, \quad (\text{R6})$$

with \mathbf{I} being the identity matrix and ϑ_1 and ϑ_2 being rotation angles

$$\vartheta_1 = \left(1 - \frac{\alpha}{\pi}\right) \cdot \frac{\pi}{2} \cdot \min\left(\frac{\rho}{r}, 1\right), \quad \vartheta_2 = \frac{\alpha}{\pi} \cdot \frac{\pi}{4} \cdot \max\left(-1, \min\left(\frac{x'}{r}, 1\right)\right) \cdot \max\left(-1, \min\left(\frac{z'}{r}, 1\right)\right), \quad (\text{R7})$$

around the following vectors

$$\mathbf{k}_1 = \begin{pmatrix} \cos(-\varphi) \\ \sin(-\varphi) \\ 0 \end{pmatrix}, \quad \mathbf{k}_2 = \begin{pmatrix} 0 \\ 1 \\ 0 \end{pmatrix}, \quad (\text{R8})$$

which are set up for the cylindrical coordinate system $\{\rho, \varphi, z'\}$: $\rho \equiv \sqrt{x'^2 + y'^2}$ and $\varphi \equiv \arg(x' + iy')$. Thus, expression (R5) is a composition of two rotation matrices, each of which is intermediate between the trivial one and another rotation matrix corresponding to either the bulk vortex state or the surface vortex state.

Comment 3:

If the stable energetic configuration for a resulting winding number is 90 degrees only, isn't it surprising that the fabricated structure presents that state?

Answer:

For the ideal tetrapod geometry with the same radii of line segments, the transition between the bulk and surface antivortices occurs at the rotation angle $\alpha = 90^\circ$ due to the T_d symmetry of the object, see Fig. R4i. In the case of an asymmetric tetrapod geometry with top and bottom line segments being of different radii $r = 40$ nm and $r = 58$ nm, respectively, energies of the bulk and surface antivortex states become very close to each other in a finite range of angles $80^\circ \leq \alpha \leq 100^\circ$, see Fig. R4j. This type of asymmetric geometry is of advantage for experimental explorations aiming to realize surface antivortex textures.

The main text has been modified accordingly (pages 2–3):

Only for $\alpha = 90^\circ$, states with the surface and bulk antivortices become symmetry-connected and, accordingly, equal in energy, while for all other α the bulk antivortex has slightly lower energy. An increase of the angular range where both these states have almost the same energies ($80^\circ \lesssim \alpha \lesssim 100^\circ$) can be achieved if the top and bottom tetrapod parts are asymmetric, e.g. the diameter of top and bottom line segments is different, see Fig. R4j. This type of asymmetric geometry is of advantage for experimental explorations aiming to realize surface antivortex textures. This enables lower magnetization gradients in the connection regions of the tetrapod line segments and favors the internal part of the connection region to be homogeneously magnetized.

Comment 4:

Has any magnetic protocol been realised to reach this state? Authors should comment about it.

Answer:

The XMCD-PEEM imaging of tetrapods was done at remanence in the as-grown state (i.e., the samples were not exposed to magnetic field prior to imaging) and after AC demagnetization in an in-plane magnetic field of 100 mT. The comparison of the magnetic states of tetrapods before and after AC demagnetization indicates that this particular field treatment was not able to affect the magnetization state of the tetrapods. Every magnetic state shown for the experimental tetrapod geometries is taken of the samples at remanence after the AC demagnetization.

The main text is modified accordingly in the section **Topologically non-trivial surface states in tetrapods** (page 3):

The magnetic states are visualized at remanence by means of XMCD-PEEM through shadow contrast imaging at the L_3 absorption edge of Co after an in-plane AC demagnetization procedure with a maximum magnetic field of 100 mT.

Comment 5:

In page 3 when discussing the possibilities to use this structures for reservoir/probabilistic computing,

I would appreciate a clarification about how the complexity necessary for these approaches could be achieved in a system so deterministic due to its well defined geometry and texture localization and stability. In fact, although the number of textures is as the author's state virtually infinite, you need to add as many rods and crosslinks that you need in order to achieve a large enough base of defects to perform reservoir computing. Moreover, how is the size of the knot affecting the formed textures as well as their location/magnetic connection is not clear.

Answer:

We thank the Reviewer for his/her valuable remark. To address the question of how connection knot size affects the magnetic texture, we performed additional micromagnetic simulations, which are now added to the revised manuscript. In particular, we substantially extended the section **Impact of the tetrapod geometry on magnetic textures** in the supplementary information (page 4):

To investigate the influence of geometric characteristics of a tetrapod on the resulting equilibrium magnetic states, we performed simulations varying geometric parameters including the rotation angle, opening angle, line segment length and its diameter.

Rotation angle between top and bottom tetrapod segments

We define α as the rotation angle between the top and bottom parts of the tetrapod. Here, we consider values of α from 0° to 180° with a step of 10° . For each case, the geometry is constructed from rounded $1.3 \mu\text{m}$ long straight nanowire segments with a radius of 57 nm and opening angle of 90° between linear segments of its top and bottom parts. Examples of tetrapod geometries which are used in micromagnetic simulations are shown in Supplementary Fig. R6. For instance, the tetrapod with the rotation angle $\alpha = 0^\circ$ stabilizes at equilibrium a bulk antivortex state with a Bloch line going through the connection area of the four line segments. The surface antivortex state is a metastable state for all geometries but $\alpha = 90^\circ$, where it possesses the same energy as the bulk one. The resulting total vorticity $Q^\Sigma = +2$ for all cases.

Opening angle

The opening angle β is defined as the angle between the bottom segments of the tetrapod; the angle between the top segments of the tetrapod is kept the same. We perform micromagnetic simulations for the tetrapod geometries with the rotation angle $\alpha = 90^\circ$ and constructed from the $1.3 \mu\text{m}$ long linear nanowires with the radius of 57 nm and opening angles $\beta \in [30^\circ; 45^\circ; 60^\circ; 75^\circ; 90^\circ]$. Starting from different initial configurations, we stabilize various equilibrium magnetic textures presented in Supplementary Fig. R2. In the case of opening angle $\beta = 30^\circ$, the equilibrium state acquires the total vorticity $Q^\Sigma = +2$, with four vortices and two antivortices. An additional shape anisotropy that appears in the central tetrapod region, due to the spatial elongation, leads to the formation of a hybrid vortex-antivortex state with two antivortices at the surface being positioned along the \hat{z} -axis, and a bulk vortex Bloch line inside the central region. For geometries with the opening angle β larger than 45° , both antivortex states are positioned in the $\hat{x}\hat{y}$ plane of the central tetrapod region.

Figure R6. The tetrapod geometry with bulk and surface antivortex states. Panels show the equilibrium magnetic state of tetrapods with different azimuthal rotation angles α of 0° , 30° , 60° and 90° between the planes accommodating the top and bottom parts of the tetrapod. Green regions in each geometry show the topological charge density distribution.

Length of the tetrapod line segments

To investigate the influence of shortening of the tetrapod line segments on the equilibrium magnetic states, we perform micromagnetic simulations for geometries constructed from rounded linear nanowires of different length, $L \in [300; 500; 700; 900; 1100; 1300]$ nm, with the same radius $r = 57$ nm, the rotation angle $\alpha = 90^\circ$ and opening angle $\beta = 90^\circ$. In the case of the tetrapod with the shortest line segments ($L = 300$ nm), the resulting equilibrium magnetic state possesses an additional vortex-antivortex pair in the central region due to the formation of the bulk vortex Bloch lines that intertwine in the connection area, see Supplementary Fig. R1(a). Thus, the surface magnetization distribution acquires 5 vortices and 3 antivortices, resulting in the total vorticity $Q^\Sigma = +2$ being in line with the Euler characteristic of the geometry $\chi = +2$, see Supplementary Fig. R1(c). With the elongation of the line segments to $L = 700$ nm, the vortex structure appears only as a surface state at their ends without the formation of Bloch lines along segments, see Supplementary Fig. R1(b). This results in the emergence of a pair of antivortices at the surface of the connection region despite the magnetization curling in linear segments near the connection area, see Supplementary Fig. R1(e). Further elongation of the line segments to $L = 1.1 \mu\text{m}$ does not result in a change of the total number of magnetic textures in the tetrapod, but it leads to the formation of homogeneously magnetized linear segments in the central area of the tetrapod with a reduction of the curling of magnetization at the surface, see Supplementary Fig. R1(c) and (f). We note that the segment length determines the strength of the interaction between vortices at the segment ends and antivortices in the central part of the geometry. If the segments are short enough, the vortex cores can substantially affect antivortices in the center, even leading to the creation of an additional vortex-antivortex pair in equilibrium.

Radius of line segments

To study the influence of the radius of line segments forming the tetrapod geometry on the equilibrium magnetic states, we construct tetrapods with $L = 1.3 \mu\text{m}$ long line segments, the opening angle $\beta = 90^\circ$ and the rotation angle $\alpha = 90^\circ$, but with different nanowire radii $r \in [10; 20; 30; 40; 50; 60; 70; 80; 90; 100]$ nm. For all constructed geometries, we observe the formation of antivortex states in the central region of the tetrapod at equilibrium, see Supplementary Fig. R3. In the case of the tetrapod constructed from narrow line segments with $r \leq 30$ nm, the line segments are homogeneously magnetized. The topological defect in the magnetic texture is presented by the out-of-surface magnetization in the flower state rather than the classical vortex distribution. Still, this state topologically equivalent to the vortex one as it is shown by its angular projection of the surface magnetization on the elemental sphere, see Fig. R3f. The increase of the radius of the line segments ($40 \leq r \leq 70$ nm) leads to the formation of a surface vortex distribution at their ends. For the case of line segments with $r \geq 80$ nm, the surface vortices fill the entire volume of nanowires with the vortex Bloch line. The formation of the bulk vortex Bloch lines may potentially introduce additional vortex-antivortex surface pairs in the central region of the tetrapod.

Furthermore, stimulated by the remark of the Reviewer, we extended the discussion on the possibility to realize physical reservoirs based on magnetic wireframes. In this respect, we note that the system should fulfil the following requirements on (i) interconnection complexity, (ii) reproducibility of reservoir states and (iii) non-linear interaction of system components.⁶ To this end,

(i) FEBID offers possibilities to realize complex wireframe structures as it was demonstrated, e.g., for the buckyball geometry.⁷ This object should host at least 60 antivortices.

(ii) The reproducibility of the state is determined by the link between the total vorticity of the surface magnetization, Q^{Σ} , and Euler characteristic, χ , of the geometry of arbitrary complexity. Namely, the position and type of the texture are primarily determined by the geometric features like segment end, which always hosts surface vortices, or joint regions, which favours antivortices.

(iii) The interaction behaviour between these topological textures can be tuned by selection of the length and diameter of the wireframe's segments, which will enrich their non-linear spatio-temporal dynamics due to the magnetostatic interaction between topological textures. Moreover, the formation of magnetic state with only antivortex textures may be beneficial as it makes impossible texture annihilation which ensures the fading memory criteria of the physical reservoir.

Hence, although we do not provide an in depth analysis of the reservoir performance of these objects (which is outside the scope of this paper), the reasoning above allows us to anticipate that these magnetic wireframes can be of use for novel computing concepts, in particular for the realization of physical magnetic reservoirs.

The following text is added to the section **Topologically non-trivial surface states in tetrapods** in the main text (page 9):

Magnetic wireframes hosting a large number of solitons could be considered as a platform for the realization of physical magnetic reservoirs for neuromorphic computing as they fulfil requirements on (i) interconnection complexity, (ii) reproducibility of reservoir states and (iii) non-linear interaction of system components.⁶ Namely, FEBID offers possibilities to realize complex wireframe structures as it was demonstrated, e.g., for the buckyball geometry⁷ supporting at least 60 antivortices. The reproducibility of the state is assured by the link between the total vorticity of the surface magnetization and the Euler characteristic of the geometry. The interaction strength can be tuned by selection of the length and diameter of the wireframe's segments, which will enrich their non-linear spatio-temporal dynamics. The formation of magnetic states with only antivortex textures may be beneficial as it prevents texture annihilation, which ensures the fading memory criteria of the physical reservoir. The direct integration of nanofabricated complex 3D wireframes into standard 2D lithographically created systems⁸ with coplanar or Ω -shaped antennas or detectors should allow extending unconventional computing into 3D, offering additional functionalities such as a higher degree of interconnectivity.

Comment 6:

In page 5, the authors state that they are using for their micromagnetic simulations the experimentally reconstructed geometry of the tetrapod but no indication on how they have achieved this is mentioned neither in the methods, nor in the main text of the manuscript. It is only mentioned that high resolution SEM images were used. It would be good to indicate if this has been performed using any kind of 3D structural reconstruction method such as photogrammetry and add the adequate references for the method/code used. I would appreciate also a comment about how the authors have addressed the issue of image alignment, if it is photogrammetry, when dealing with FEBID structures with almost no features in

the surface of the structure.

Answer:

The discussed tetrapod geometry is one of the simplest 3D wireframes, which contains only four linear segments connected in one point. Thus, the experimental tetrapod was imaged by means of high-resolution SEM at different angles. These experimental data provided information on spatial dimensions, shape features and aspect ratios of the experimental geometry. In this respect, we benefit from the basic principles of the photogrammetry approach and finalize reconstruction by fitting the geometric shape with projections using sculpturing procedure in the Blender software.

We added the following explanation to the **Micromagnetics** section in the main text (page 15):

The experimental geometry is reconstructed from a series of high-resolution scanning electron microscopy images obtained for different azimuthal angles, see Fig. 2b–d, based on the sculpturing method in the Blender software with all spatial features of the tetrapod reflected in the final geometry. The resulting mesh is constructed from tetrahedron elements with an average size of 4.9 nm and volume of 27.2 nm³.

Comment 7:

Also in page 5, authors make the discussion about the connection between the two surface antivortices based on the absence of a Bloch line connecting their cores. This is also the approach followed for the rest of analysed textures. Wouldn't be interesting to show this connection/isolation by representing/analysing the magnetic vorticity/gyrovector/magnetic emergent field of the magnetization? This is a very powerful method that allows to observe how different topologically interesting items within a magnetic system are linked as previously used in "C. Donnelly et al Nature Physics 17, pages 316–321 (2021)" or in "J. Hermosa et al Communications Physics 6, 49 (2023)". This representation could be excellent for the applications indicated by the authors related with magnonics and spin-wave propagation through domain walls and Bloch lines.

Answer:

We thank the Reviewer for this suggestion. In our discussion, we decided to focus on the analysis of the distribution of the flux density of the topological charge^{9–12} over the 3D geometry:

$$\Omega_l = \frac{1}{8\pi} \varepsilon_{lno} \varepsilon_{ijk} m_i \partial_n m_j \partial_o m_k \equiv \frac{1}{8\pi} \varepsilon_{lno} \mathbf{m} \cdot [\partial_n \mathbf{m} \times \partial_o \mathbf{m}], \quad (\text{R9})$$

where m_i is the normalized local magnetization component, ε_{ijk} is the Levi–Civita tensor and $i, j, k, l, n, o = \{x, y, z\}$. Eq. (R9) is equivalent to the definition of the components of the emergent field $B_i^e = \frac{\hbar}{2} \varepsilon_{ijk} \mathbf{m} \cdot [\partial_j \mathbf{m} \times \partial_k \mathbf{m}]$ ^{13,14} up to the scaling multiplier. Thus, there are two approaches to localize topologically non-trivial magnetic textures based on the analysis of $\mathbf{\Omega}$ or \mathbf{B}^e : (i) identify maxima on the distribution of the topological charge density, $\Omega = |\mathbf{\Omega}|$ as done in this work (see also Refs.^{12,15}) or (ii) identify regions with high density of the streamlines of \mathbf{B}^e .^{12,14}

Following the suggestion of the Reviewer, we checked the distribution of an emergent field \mathbf{B}^e inside the tetrapod and compared it with analysis performed in our manuscript, see Fig. R7. To our subjective view, it seems to be easier to localize magnetic textures when following the maxima on the distribution of

Figure R7. Topological flux density and emergent field in the tetrapod. (a) The experimental tetrapod geometry with isosurfaces in the distribution of the topological charge density $\bar{\Omega} = 0.03$. Maxima in these distributions correspond to the locations of topologically non-trivial textures. (b) The distribution of the emergent field B^e inside the tetrapod geometry being depicted as streamlines. The region with a high density of lines indicates on the appearance of topologically non-trivial textures. (c) Streamlines of B^e originating from the surface non-trivial magnetic states at the ends of line segments and in the connection region of the tetrapod.

the topological charge density, see Fig. R7a. The approach based on the analysis of streamlines of the emergent field, see Fig. R7b, benefits from an additional processing to get access to the surface states, see Fig. R7c.

The following text is added to the **Methods** section of the main text (page 16):

The here applied analysis approach could be extended by the study of emergent magnetic fields,¹⁴ $B^e = 4\pi\hbar\Omega$. This method is suitable for the calculation of the Berry phase and corresponding topological Hall effect.¹³

Comment 8:

Imaging conditions of the PEEM are not fully indicated in the methods. I mean the acceleration voltage for the extraction of photoelectrons. Was it 10 keV or 20 keV or other? How this affected the measurements in terms of discharges/resolution conditions?

Answer:

We apologise for this incomplete information provided in the initial submission. The experiment was done at 10 keV to minimize the risk of discharges. Reducing the extraction voltage to 10 keV lowers the spatial resolution by a factor of 1.4, which was partially compensated by the better energy filtering and therefore lower chromatic aberration. Still, a spatial resolution of down to 30 nm was demonstrated in this imaging mode.

We added these measurement details to the **XMCD-PEEM measurements** section in the methods of the revised manuscript (page 14):

The experiment is done at 10 keV to minimize the risk of discharges. Reducing the extraction voltage from 20 keV to 10 keV lowers the spatial resolution by a factor of 1.4, which is partially compensated by the better energy filtering and therefore lower chromatic aberration. Thus, the resulting spatial resolution of down to 30 nm is achieved in this imaging mode.

Comment 9:

In figure 4, although the structures are claimed to be fabricated on top of a non-magnetic pillar, there is XMCD signal in the base of the structures showing also a vortex-compatible signal. Could the authors comment about those features and their potential impact on the final magnetic state of the system?

Answer:

We thank the Reviewer for this question. The additional dipolar XMCD-PEEM contrast on supporting non-magnetic PtC pillar originates from the co-deposited Co_3Fe layer, which is inevitably occurring upon FEBID preparation. The present red-white-blue contrast indicates on the formation of a circulating magnetic distribution within the Co_3Fe coating on the pillar. Still, this co-deposited magnetic layer does not change the resulting Euler characteristic of the magnetic geometry. Indeed, this additional coating does not lead to the formation of holes in the system. Thus, its influence on the overall topological picture of the wireframe geometry is negligible.

The following discussion has been added to the section **Design of high-order vorticity states** in the main text (page 8):

These non-magnetic PtC pillars are covered with a layer of Co_3Fe due to its co-deposition upon FEBID preparation of the magnetic wireframes. Still, the magnetic coating on the pillar does not change the resulting Euler characteristic of the magnetic geometry. Indeed, this additional coating does not lead to the formation of holes in the system. Thus, its influence on the overall topological picture of the wireframe geometry is negligible. We note that all structures are imaged in the as-grown state without exposing the samples to magnetic fields before imaging.

Comment 10:

Again about these more complex and N-torus like structures, how are the magnetic states imaged obtained? Are the samples in the as-grown state?

Answer:

The magnetic states were imaged by means of XMCD-PEEM similar to the initially discussed tetrapod geometries. The samples were imaged in the as-grown state (i.e., without any magnetic field treatment prior to imaging) at the acceleration voltage of 10 keV to minimize the risk of discharges.

The following discussion has been added to the **XMCD-PEEM measurements** section in the methods of the main text of the manuscript (page 14):

The experiment is done at 10 keV to minimize the risk of discharges. Reducing the extraction voltage from 20 keV to 10 keV lowers the spatial resolution by a factor of 1.4, which is partially compensated by the better energy filtering and therefore lower chromatic aberration. Thus, the resulting spatial resolution of down to 30 nm is achieved in this imaging mode.

and to the section **Design of high-order vorticity states** in the main text (page 8):

We note that all structures are imaged in the as-grown state without exposing the samples to magnetic fields before imaging.

Comment 11:

Could the magnetic state present in the samples be controlled by the fabrication order of the rods of the wireframe due to magnetostatic interactions between the rods?

Answer:

Upon fabrication of a wireframe structure, the magnetostatic interaction may influence the magnetic state. For example, the magnetization direction in a newly written line segment can be affected by the magnetization direction in previously written segments. Still the final total vorticity will not be affected as this is the property of the final sample geometry.

Comment 12:

In figure 5, the panel a) is under my point of view somehow confusing. Here, the streamlines following the stray field of the structure have a rainbow colormap with associated to the B_z component intensity but the colormap scale is missing. Although one can reconstruct its meaning from the magnetic configuration in the tetrapod, I think it would be easier for the reader to have the colormap scale pointing out what means positive and negative, as well as more and less intense.

Answer:

Following the suggestion of the Reviewer, we modified Figure 5a by adding color maps of the distributions of m_z and B_z/B_z^{\max} . For convenience, the changes can be seen in Figure R8.

Comment 13:

The discussion about the so called “Complex stray field texture emerging from the anti-vortices” is quite obscure as it is not clearly observed from the central cross section representation. This is I think because the authors want to explain the rotation of the field due to the geometry of the tetrapod at the same time that they point out this field texture around the antivortices at the connection. I would separate both discussions and magnify the area around the antivortices to favour the identification, discussion and understanding of this mentioned texture.

Answer:

Following the suggestion of the Reviewer, we updated Figure 5 by adding a new panel (b) showing the magnetic near-field distribution around the central area of the tetrapod, see Figure R8b.

The following discussion is added to the main text of the manuscript, section **Topology of magnetic field nanotextures** (page 9):

Figure R8. Near-field magnetic field textures around the magnetic tetrapod. (a) Streamlines of the magnetic field that have been obtained from the reconstructed magnetic state of the FEBID-grown tetrapod (Fig. 2h). The red-white-blue contrast depicts the magnetization distribution over the tetrapod surface, while the rainbow contrast represents the B_z component of the stray-field distribution. (b) Streamlines of the magnetic near-field originating from two surface antivortices in the central area of the tetrapod. Green regions show isosurfaces of the topological charge density $\tilde{\Omega} = 0.03$. Families of (c–e) vertical and (f–h) horizontal cross-sections of the calculated distribution of the magnetic field texture. Colour of individual arrows depicts the direction of the B_z component. Isosurfaces of B_z/B_z^{\max} are bent due to the geometrical frustration of the experimental tetrapod geometry. The isosurfaces are shown for (i) $B_z/B_z^{\max} = 0$, (j) $B_z/B_z^{\max} = 3 \times 10^{-4}$ and (k) $B_z/B_z^{\max} = -3 \times 10^{-4}$. Further details on topological stray field nanotextures are summarized in Supplementary Section 6.

Moreover, the distribution of the magnetic near-field around the central connection region of the tetrapod reveals a highly divergent field profile originating from the surface antivortices in that area, see Figs. R8b. Namely, each surface texture acts as a source of a strong magnetic near field, which bends into loops when moving away from the central area. These high-gradient magnetic fields may be potentially utilized for pinning and guiding of ultracold atoms by means of non-linear magnetic fields.¹⁶

Comment 14:

To finish with figure 5, the solid colours selected for the representation of the B_z/B_z^{\max} isosurfaces are confusing in terms of the rainbow colormap which appears in figure 5 panel k). The colours of the isosurfaces does not correspond to the levels shown in the colormap scale.

Answer:

Following the suggestion of the Reviewer, we adjusted colours in panels (i–k) of Figure 5 of the main text. For convenience, the changes can be seen in Figure R8.

References

1. Kravchuk, V. P. *et al.* Topologically stable magnetization states on a spherical shell: Curvature-stabilized skyrmions. *Physical Review B* **94**, 144402 (2016). URL <http://link.aps.org/doi/10.1103/PhysRevB.94.144402>.
2. Kravchuk, V. P., Sheka, D. D., Rößler, U. K., van den Brink, J. & Gaididei, Y. Spin eigenmodes of magnetic skyrmions and the problem of the effective skyrmion mass. *Physical Review B* **97**, 064403 (2018). URL <https://link.aps.org/doi/10.1103/PhysRevB.97.064403>.
3. Porrati, F. *et al.* Site-selective chemical vapor deposition on direct-write 3d nanoarchitectures. *ACS Nano* (2023).
4. Kravchuk, V. P. *et al.* Multiplet of skyrmion states on a curvilinear defect: Reconfigurable skyrmion lattices. *Physical Review Letters* **120**, 067201 (2018). URL <https://link.aps.org/doi/10.1103/PhysRevLett.120.067201>.
5. Hubert, A. & Schäfer, R. *Magnetic domains: The analysis of magnetic microstructures* (Springer Berlin Heidelberg, Berlin, 2009). URL <https://doi.org/10.1007/978-3-540-85054-0>.
6. Cucchi, M., Abreu, S., Ciccone, G., Brunner, D. & Kleemann, H. Hands-on reservoir computing: a tutorial for practical implementation. *Neuromorphic Computing and Engineering* **2**, 032002 (2022).
7. Keller, L. & Huth, M. Pattern generation for direct-write three-dimensional nanoscale structures via focused electron beam induced deposition. *Beilstein J. Nanotechnol.* **9**, 2581–2598 (2018). URL <https://doi.org/10.3762/bjnano.9.240>.
8. Meng, F. *et al.* Non-planar geometrical effects on the magnetoelectrical signal in a three-dimensional nanomagnetic circuit. *ACS Nano* **15**, 6765–6773 (2021).
9. Papanicolaou, N. & Tomaras, T. N. Dynamics of magnetic vortices. *Nuclear Physics B* **360**, 425–462 (1991). URL [https://doi.org/10.1016/0550-3213\(91\)90410-Y](https://doi.org/10.1016/0550-3213(91)90410-Y).

10. Papanicolaou, N. Dynamics of magnetic vortex rings. In *Singularities in fluids, plasmas, and optics*, vol. 404 (Springer Netherlands, 1993). URL https://www.ebook.de/de/product/21700433/singularities_in_fluids_plasmas_and_optics.html.
11. Cooper, N. R. Propagating magnetic vortex rings in ferromagnets. *Physical Review Letters* **82**, 1554–1557 (1999). URL <http://link.aps.org/doi/10.1103/PhysRevLett.82.1554>.
12. Donnelly, C. *et al.* Experimental observation of vortex rings in a bulk magnet. *Nature Physics* **17**, 316–321 (2020). URL <https://www.nature.com/articles/s41567-020-01057-3>.
13. Bruno, P., Dugaev, V. K. & Taillefumier, M. Topological Hall effect and Berry phase in magnetic nanostructures. *Physical Review Letters* **93**, 096806 (2004). URL <http://dx.doi.org/10.1103/PhysRevLett.93.096806>.
14. Hermosa, J. *et al.* Bloch points and topological dipoles observed by x-ray vector magnetic tomography in a ferromagnetic microstructure. *Communications Physics* **6** (2023).
15. Volkov, O. M. *et al.* Chirality coupling in topological magnetic textures with multiple magnetochiral parameters. *Nature Communications* **14**, 1491 (2023).
16. West, A. D. *et al.* Realization of the manipulation of ultracold atoms with a reconfigurable nanomagnetic system of domain walls. *Nano Letters* **12**, 4065–4069 (2012). URL <https://doi.org/10.1021/nl301491m>.

Reviewers' Comments:

Reviewer #1:

Remarks to the Author:

I thank the authors for their detailed answers and the additional simulations. Overall, I find the answers very convincing and satisfactory. I now recommend this manuscript for publication in Nature Communications.

Reviewer #2:

Remarks to the Author:

The authors have addressed all the points raised by the reviewers, performed additional calculations and included further discussions bringing clarity to the manuscript. Due to this, I recommend the publication of the manuscript in Nature Communications in its present revised form.